# A New Backbone for Hyperspectral Image Reconstruction

## Abstract

As the inverse process of snapshot compressive imaging, the hyperspectral image (HSI) reconstruction takes the 2D measurement as input and posteriorly retrieves the captured 3D spatial-spectral signal. Built upon several assumptions, numerous sophisticated neural networks have come to the fore in this task. Despite their prosperity under experimental settings, it's still extremely challenging for existing networks to achieve high-fidelity **reconstructive quality** while maximizing the **reconstructive efficiency** (computational efficiency and power occupation), which prohibits their further deployment in practical applications. In this paper, we firstly conduct a retrospective analysis on aforementioned assumptions, through which we indicate the imminent aspiration for an authentically practical-oriented network in reconstructive community. By analysing the effectiveness and limitations of the widely-used reconstructive backbone U-Net, we propose a Simple Reconstruction Network, namely SRN, just based on some popular techniques, e.g., scale/spectral-invariant learning and identity connection. It turns out, under current conditions, such a pragmatic solution outperforms existing reconstructive methods by an obvious margin and maximize the reconstructive efficiency concretely. We hope the proposed SRN can further contribute to the cutting-edge reconstructive methods as a promising backbone, and also benefit the realistic tasks, i.e., real-time/high-resolution HSI reconstruction, solely as a baseline.

## 1 Introduction

Hyperspectral imaging (HSI) refers to multi-channel imaging where each channel stores information at a specific spectral wavelength for a fixed real-world scene (Plaza et al., 2009). By capturing spatial intensity in a spectral-wise manner, hyperspectral images empower richer information than traditional RGB image cubes and they have been applied in a wide range of scenarios, *e.g.*, object detection (Kim et al., 2012; Xu et al., 2015), remote sensing (Borengasser et al., 2007; Melgani & Bruzzone, 2004; Yuan et al., 2017), medical image processing (Lu & Fei, 2014; Meng et al., 2020c) etc. HSI can be captured and measured by snapshot compressive imaging (SCI) systems, which tend to compress information of snapshots along the spectral axis into one single 2D measurement (Yuan et al., 2021). The coded aperture snapshot spectral imaging (CASSI) system (Wagadarikar et al., 2008; Meng et al., 2020b) forms one mainstream research direction among existing SCI systems due to its passive modulation property (Llull et al., 2013; Wagadarikar et al., 2008; 2009; Yuan et al., 2015).

The goal of *HSI reconstruction* is to transform the measurements into desired cubic hyperspectral images. As a result, a dimensional-expansion mapping function (2D to 3D) is required, for which reason such a mapping relationship approximation is deemed to be much harder than general image regression tasks. By introducing domain expertise, previous research efforts have proposed a quite number of reconstruction algorithms (Bioucas-Dias & Figueiredo, 2007; Liu et al., 2019; Miao et al., 2019; Meng et al., 2020b; Wang et al., 2020; 2017; 2019; Yuan, 2016), among which deep neural networks (Meng et al., 2020b;c; Miao et al., 2019; Wang et al., 2019; Wang et al., 2019; 2020; Zheng et al., 2021) enable an effective way to faithfully bridge between input and output compositional hierarchies (LeCun et al., 2015).

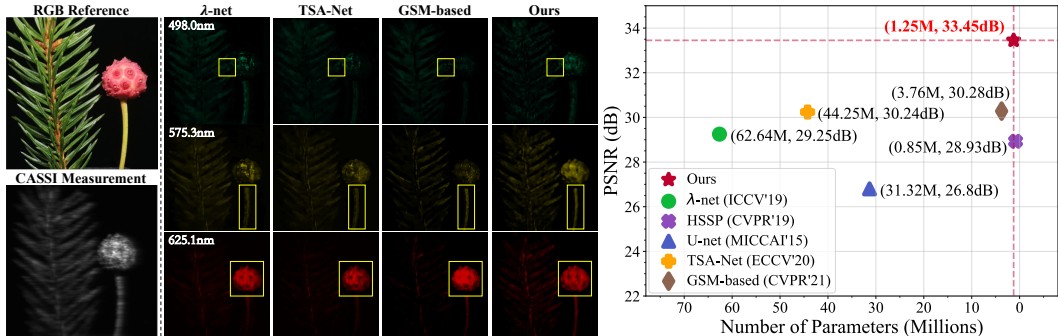

Figure 1: *Left*: HSI reconstructions between the proposed SRN and several state-of-the-art methods. The RGB reference (top-left) is shown to demonstrate the color. The reconstructed details are highlighted with yellow bounding boxes (zoom in for a clear comparison). *Right*: Comparison between different HSI reconstruction methods in terms of PSNR ($\uparrow$) and model size ($\downarrow$).

## 1.1 MOTIVATION

While neural networks become increasingly prevalent in solving the HSI reconstruction problem, their success is reliant on several idealistic hypotheses, which hardly holds in practical scenarios. This prohibits the further application and exploration of HSI.

**Proper Dataset Volume.** The feasibility of learning in a universal sense can be defined by satisfying a VC generalization bound Abu-Mostafa et al. (2012). From this perspective, deep neural networks are considered to be high potential due to their high VC bound (which can be described as the amount of free variables). Notably, the corresponding feasibility of neural networks is generally determined by their model complexity and the volume of datasets. In the recent HSI reconstruction methods, more increasingly complicated reconstructive networks have been proposed by assuming the underlying dataset is sufficiently large so that the learning is feasible. This neglects the fact that there's limited data accessible. Under such circumstance, simpler model might be more promising.

**Ideal reconstructive rate.** Practically, any reconstructive algorithm appears with a substantial forward model. For temporal-insensitive situations, the inverse model processes the measurements at any reconstructive rate $r_{recon} > 0$, without considering cooperating with the forward model. However, the reconstructive rate should be higher than (or at least equal to) capture rate of the forward model, i.e., $r_{recon} \geq r_{cap} > 0$, for temporal-sensitive cases, i.e., real-time/high-speed HSI reconstruction. For example, $\lambda$-net proposed by Miao et al. (2019) can be applied in reality with $r_{recon} > 30fps$. More generally, reconstructive methods are always hypothetically deemed to be efficient enough, i.e., compatible with any forward systems, which however probably suffers from a deviation from current technical support. Considering the arbitrariness of the lower bound $r_{cap}$, we cannot determine an "ideal" reconstructive rate but only to maximize it, i.e., minimize the reconstructive time. For neural network-based methods, eliminating the number of parameters and computational burdens is a sound solution.

**Unconstrained computing power.** The success of the deep neural networks can largely attribute to the overparameterization (Soltanolkotabi et al., 2018)—the parameters in neural networks are a lot more than the training samples for a good representation learning. Actually, just by investing more computing power $P$, the cornerstone which overparametrization is reliant upon, researchers have made great progress in diverse applications (Thompson et al., 2020). Latent performance boost can be further expected if more computing power is available, until one reaches the computing power bottleneck $P^*$. It turns out it's difficult to solve such a bottleneck in practical. For example, to conduct TSA-Net(Meng et al., 2020b) for high resolution (HR)-HSI restoration (1024×1024), the required GPU memory would be unrestrictedly over 18Gb, which indicates dilemma encountered by all high-performanced reconstructive methods with large model size.

## 1.2 CONTRIBUTIONS

In this study, bearing the limitations imposed by the scale of datasets, the capability of optical systems, and the computational platform, we posit the possibility of network construction with min-

imum inductive bias, based on an observation upon the reconstructive baseline U-net. We propose a simple yet quite promising CNN reconstructive network, namely Simple Recon Net (SRN), whose success is owing to revisiting and tailoring several practical techniques for HSI, i.e., spatial/spectral invariant learning and residual learning. We summarize the contribution of this work as follows.

- The proposed SRN provides a new state-of-the-art by improving the previous one (Huang et al., 2021) $> 3dB$ in PSNR. Moreover, our approach presents clear perceptual improvement across different spectral channels (Fig. 1 *Left*).
- We significantly shrink the model size (see Fig. 1 *Right*) without sacrificing the performance, i.e., to most extent, we only use $<1/3$ parameters of GSM-based method (Huang et al., 2021) and to the maximum, reduce FLOPs by $> 34 times$ as shown in Tab. 3, both of which yield a higher temporal efficiency and computational efficiency. This makes real-time/higher-resolution images reconstruction practically executable under restricted GPU conditions.
- The proposed network can either be solely used as a baseline, or conjunctively referred to as a backbone, contributing to more complicated methods, i,e., more sophisticated E2E methods or unrolling methods.

## 2 RELATED WORKS

The basic idea of SCI is to modulate the high-dimensional signal using a higher frequency than the capture rate of the camera. In this manner, a compressed coded frame obtained will include the information in the high-dimensional signal and a high-performance algorithm can then be employed to recover the desired data. For compression, the novel implementation of SCI, CASSI, uses a coded aperture and a prism to conduct the (spatio-)spectral modulation.

Previously, iterative-based optimization algorithms predominate the field of HSI reconstruction by approximating the image priors through diverse regularization techniques, i.e., the total variance (TV) (Kittle et al., 2010; Wang et al., 2015; Wagadarikar et al., 2008), sparsity (Wang et al., 2017; 2015), non-local similarity (NLS) (Wang et al., 2016; Fu et al., 2016), Markov Random Field (MRF) (Tappen, 2007), Gaussian mixture model (Yang et al., 2015) etc, among which De-SCI (Liu et al., 2019) achieves best performance on both video and spectral compressive imaging. TwIST (Bioucas-Dias & Figueiredo, 2007) proposed a two-step Iterative shrinkage/thresholding algorithm by modeling the reconstructive problem as a a linear observation model with a nonquadratic regularizer (i.e., total variation). GPSR (M. A. T. Figueiredo et al., 2007) proposes to use gradient projection (GP) algorithms to solve the inverse problem that is formulated as bound-constrained quadratic programming (BCQP) process.

Inspired by the success of deep learning in other image translation problems, researchers have started using deep learning to reconstruct hyperspectral images from CASSI measurements (Meng et al., 2020b;c; Miao et al., 2019; Wang et al., 2019; Wang et al., 2019; 2020; Zheng et al., 2021), which can be substantially divided into two streams: end-to-end neural networks (E2E-NN) models and others. The former stream tends to directly learn a complete mapping function from measurements (always packaged with masks) to estimated HSIs. Other researchers turns to introduce NN models into conventional optimization algorithms, named deep unrolling/unfolding nad plug-and-play methods, leading to lightweight and interpretable methods.

Proposed in Ronneberger et al. (2015), U-Net sat atop leaderboards regarding medical image tracking and segmentation, followed by which many variants have been derived in recent years, i.e., 3D U-Net (Çiçek et al., 2016), Ternaus U-Net (Iglovikov & Shvets, 2018), MultiResUNet (Ibtehaz & Rahman, 2020) and Attention U-Net (Oktay et al., 2018) etc. Typical investigation correlates such a predominance with characteristics of medical images: 1) repeated tissue patterns and complicated gradient distribution; 2) locally anomalous region and atypical noise distribution. U-Net intelligently fits the above characteristics by firstly abstracts coarse information and then focus on localization through simultaneously expansion and skip connection.

As a classical network structure, U-Net serves as the first precast backbone in HSI reconstruction, among a series of famous architectures, and has been faithfully employed in both E2E and unrolling methods. For example, the $\lambda$-net (Miao et al., 2019) is a dual-stage generative model which employs a U-Net and residual learning strategy. The TSA-Net (Meng et al., 2020b) which combined

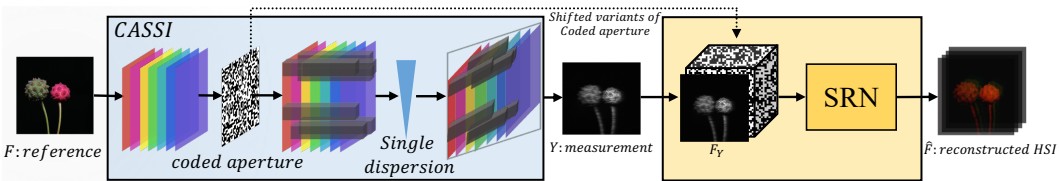

Figure 2: Illustration of HSI capturing and reconstruction. An optical system, CASSI (left blue box, in which a 2D coded aperture and a disperse are displayed while other optical components are omitted), compresses the hyperspectral signal $\boldsymbol{F}$ into a 2D measurement $\boldsymbol{Y}$. For reconstruction (right yellow box), we train the proposed Simple Recon Net (SRN), to learn an E2E mapping from $\boldsymbol{Y}$ (initialized by shifted variants of coded aperture) to $\widehat{\boldsymbol{F}}$.

spatial-spectral self-attention with U-Net led to excellent results on both simulation and real data. As a deep unfolding method, GAP-net (Meng et al., 2020a) utilizes 15-layer U-Net as a trained denoiser in each stage. Recent Gaussian Scale Mixture Prior-based (GSM-based) baseline (Huang et al., 2021) employing U-Net for two objects: a lightweight U-Net for approximating the regularization parameters, another lightweight U-Net for estimating the local-mean of GSM prior. U-Net has exemplified the rationality of uncomplicated neural network in field of HSI reconstruction, even if considering the huge difference between the medical images and hyperspectral images. Actually, as reported in Tab. 1 and Tab. 2, U-net only achieves sub-optimal performance when solely referred to as a reconstructive baseline, which reveals both the validity and the limitations of such a generic solution. It turns out the performance of neural networks are sensitive to minor adjustments. Compared with the U-Net, it's possible to construct a "substitute network" that enables a significant performance boost with mainstream techniques at hand.

## 3 METHODOLOGY

### 3.1 MATHEMATICAL MODEL OF CASSI

Recall that the key idea of SCI is to modulate the spatio-spectral signal with different modulation patterns, and CASSI implemented this in a low cost way by a fixed mask plus a disperser. Given the 3D spectral cube (Fig. 2 left), denoted by $\boldsymbol{F} \in \mathbb{R}^{N_x \times N_y \times N_\lambda}$, where $N_x$, $N_y$, and $N_\lambda$ represent the spectral image's height, width, and total number of wavelengths, respectively, we compute signal modulation (implemented by the mask) in a channel-wisely way as $\boldsymbol{F}'(:,:,n_\lambda) = \boldsymbol{F}(:,:,n_\lambda) \odot \boldsymbol{M}^*$, where $\boldsymbol{F}' \in \mathbb{R}^{N_x \times N_y \times N_\lambda}$ represents the modulated signals, $\boldsymbol{M}^* \in \mathbb{R}^{N_x \times N_y}$ refers to a pre-defined physical mask governing on signal modulation, $n_\lambda \in [1, \dots, N_\lambda]$ indexes wavelengths, and $\odot$ is the Hadamard product. By passing $\boldsymbol{F}'$ to a disperser, the cube $\boldsymbol{F}'$ becomes tilted and could be sheared along the $y$-axis. Let $\boldsymbol{F}'' \in \mathbb{R}^{N_x \times (Ny+N_\lambda-1) \times N_\lambda}$ be the tilted cube, and $\lambda_c$ be a reference wavelength, i.e., $\boldsymbol{F}'[:,:,n_{\lambda_c}]$ works like an anchor image that is not sheared along the $y$-axis, we have $\boldsymbol{F}''(u,v,n_\lambda) = \boldsymbol{F}'(x, y+d(\lambda_n-\lambda_c), n_\lambda)$, where $(u,v)$ locates the coordinate system on the detector plane, $\lambda_n$ denotes the $n_\lambda$-th channel, $\lambda_c$ refers to the anchored wavelength, and $d(\lambda_n-\lambda_c)$ represents a spatial shift of the $n_\lambda$-th channel in $\boldsymbol{F}'$. Notably, since the sensor integrates all the light within the wavelength range $[\lambda_{\min}, \lambda_{\max}]$, we could model the compressed measurement at a detector $y(u,v)$ with the following integral: $y(u,v) = \int_{\lambda_{\min}}^{\lambda_{\max}} f''(u,v,n_\lambda) \mathrm{d}\lambda$, where $f''$ is the analog (continuous) representation of $\boldsymbol{F}''$. To discretize this model, we have $\boldsymbol{Y} = \sum_{n_\lambda=1}^{N_\lambda} \boldsymbol{F}''(:,:,n_\lambda) + \boldsymbol{G}$. $\boldsymbol{Y} \in \mathbb{R}^{N_x \times (Ny+N_\lambda-1)}$ represents a 2D measurement, which, in essence, is a *compressed* frame capturing the information, and $\boldsymbol{G} \in \mathbb{R}^{N_x \times (Ny+N_\lambda-1)}$ is a corresponding 2D measurement noise.

To simplify model notations, we define $\boldsymbol{M} \in \mathbb{R}^{N_x \times (Ny+N_\lambda-1) \times N_\lambda}$ and $\tilde{\boldsymbol{F}} \in \mathbb{R}^{N_x \times (Ny+N_\lambda-1) \times N_\lambda}$ as the shifted masks and signal frames of different wavelengths as follows: $\boldsymbol{M}(u,v,n_\lambda) = \boldsymbol{M}^*(x, y + d(\lambda_n - \lambda_c))$, $\tilde{\boldsymbol{F}}(u,v,n_\lambda) = \boldsymbol{F}(x, y + d(\lambda_n - \lambda_c), n_\lambda)$. By using $\boldsymbol{M}$ and $\tilde{\boldsymbol{F}}$, the measurement $\boldsymbol{Y}$ could be reformulated as

$$\boldsymbol{Y} = \sum_{n_\lambda=1}^{N_\lambda} \tilde{\boldsymbol{F}}(:,:,n_\lambda) \odot \boldsymbol{M}(:,:,n_\lambda) + \boldsymbol{G}. \tag{1}$$

It is usually easier to use the vectorized notations. Let $\boldsymbol{y} = \mathrm{vec}(\boldsymbol{Y})$ and $\boldsymbol{g} = \mathrm{vec}(\boldsymbol{G}) \in \mathbb{R}^n$ be the vectorization of matrices $\boldsymbol{Y}$ and $\boldsymbol{G}$, where $\mathrm{vec}(\cdot)$ concatenates all the columns of a matrix

as one single vector. Similarly, we have $\tilde{\boldsymbol{f}}^{(n_\lambda)} = \text{vec}(\tilde{\boldsymbol{F}}(:,:,n_\lambda))$, resulting in the vector $\boldsymbol{f} = \text{vec}([\tilde{\boldsymbol{f}}^{(1)} \ldots \tilde{\boldsymbol{f}}^{(N_\lambda)}]) \in \mathbb{R}^{nN_\lambda}$, where $n = N_x(N_y + N_\lambda - 1)$. By defining the sensing matrix as $\boldsymbol{\Phi} = [\boldsymbol{D}_1, \ldots, \boldsymbol{D}_{N_\lambda}] \in \mathbb{R}^{n \times nN_\lambda}$, where $\boldsymbol{D}_{n_\lambda} = \text{Diag}(\text{vec}(\boldsymbol{M}(:,:,n_\lambda)))$ is a diagonal matrix expanded by $\text{vec}(\boldsymbol{M}(:,:,n_\lambda))$, rewrite equation 1 as

$$\boldsymbol{y} = \boldsymbol{\Phi}\boldsymbol{f} + \boldsymbol{g}. \tag{2}$$

While obtaining the vectorized measure $\boldsymbol{y}$ is similar to compressive sensing (Donoho, 2006; Emmanuel et al., 2006), the sensing matrix $\boldsymbol{\Phi}$ has a very special structure. The recent studies (Jalali & Yuan, 2019) have shown the signal can still be recovered even when $N_\lambda > 1$. Given the measurement $\boldsymbol{y}$ captured by the camera and $\boldsymbol{\Phi}$ calibrated upon pre-design, one critical and practical problem of CASSI is to solve $\boldsymbol{f}$ used in equation 2, falling in the vein of inverse problem (Yuan et al., 2021), for which we will propose a simple yet promising deep convolutional neural network below.

## 3.2 NETWORK DESIGN

The reconstructive function generates the 3D hyperspectral estimations from measurements. Therefore, it would be better if one decompose the 2D measurements into 3D data cube beforehand. A typical operation is to firstly initialize network input by conducting dot productions between the measurement and each spectral channels of shifted variants of coded aperture, namely mask, with an intention to mimic the $\boldsymbol{\Phi}^{-1}\boldsymbol{y}$ according to equation 2:

$$\boldsymbol{F}_Y[:,:,n_\lambda] := shift(\boldsymbol{M}_{n_\lambda} \odot \boldsymbol{Y}), \tag{3}$$

where $n_\lambda$-th channel of $\boldsymbol{F}_Y \in \mathbb{R}^{N_x \times Ny \times N_\lambda}$ is computed through Hadamard product, i.e., $\odot$, and $shift$ operation (raw output will be shifted into the desired tensor shape). Overall, the proposed model aims to learn the mapping as $\boldsymbol{G}(\cdot) : \boldsymbol{F}_Y \to \widehat{\boldsymbol{F}}$, where $\widehat{\boldsymbol{F}}$ denotes the reconstruction result. It has been previously proved that network depth is one of the determining factors for the success of estimation (Liang & Srikant, 2016; Telgarsky, 2016; Eldan & Shamir, 2016). We thereby preclude complicated modules while maintain sufficient depth. The specific network structure is quite simple and have the following function format:

$$g_k = \begin{cases} g_0, k = 0 \\ g_0 + \sum_{i=1}^{k} \boldsymbol{R}_i(g_{i-1}), k = 1, \ldots, K, \end{cases} \tag{4}$$

where $g_k$ is intermediate embedding at level $k$, $g_0$ is initialized by a single convolutional layer, i.e., $g_0 = \text{CONV}(\boldsymbol{F}_Y)$, where $g_0 \in \mathbb{R}^{N_x \times Ny \times N_\gamma}$. Note that $N_\gamma > N_\lambda$, based on which we intend to do a "learnable spectral interpolation".

This provides more spectral redundancy and thus enables the model to better learn the connections among different channels. For example, adjacent spectral channels tend to be more correlated to each other in general HSIs. $\boldsymbol{R}_k(\cdot)$ in equation 4 is just a non-linear feature transformer that bridges between different levels of embedding. We simply set it as $\boldsymbol{R}_k(x) = \text{CONV}(\text{ReLU}(\text{CONV}(x)))$. As shown in Fig. 3, the main structure of the proposed network includes $K$ different levels of embedding. Symmetric with the input initialization,

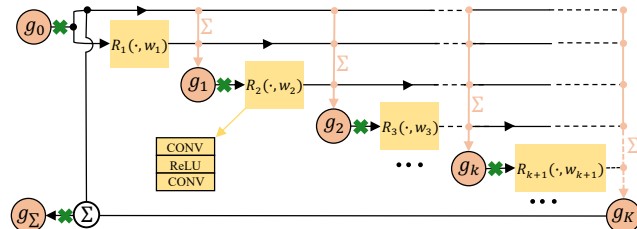

Figure 3: Illustration of the proposed Simple Recon Net (SRN). Input $g_0 = \text{CONV}(\boldsymbol{F}_Y)$. All the $\boldsymbol{R}_k(\cdot), k = 1, \ldots, K$, are defined by CONV-RELU-CONV. To maximize efficiency, rescaling pairs are inserted right after the intermediate embeddings, indicated by $\times$. For example, in SRN v2, we do downsample for $g_4$ and then upsample for $g_{12}$.

the final output is simply computed by a convolutional layer: $\widehat{\boldsymbol{F}} = \text{CONV}(g_\Sigma) = \text{CONV}(g_K + g_0)$. Identity connections (Bishop et al., 1995) are widely used in the whole network. In Sec. 4, we show that such a simple network stands out among all existing reconstructive methods. Below we will introduce several guidelines for model establishment, followed by which the proposed network successfully learns the mapping function between cubic input-output data pairs.

### 3.2.1 SPATIAL/SPECTRAL INVARIANCE

As we mentioned in Sec. 2, U-net (Ronneberger et al., 2015) is barely satisfactory in this field, even being technically modified as in Miao et al. (2019). This is determined by twofold reasons: 1) Different characteristic between medical images and HSIs (natural scenes) as illustrated in Sec. 2; 2) Different utilities: medical image processing task is always more locally focused while the HSI reconstruction tends to be globally comprehensive.

It turns out the receptive field (RF) is important for image tasks. As demonstrated by Luo et al. (2016), for plenty of tasks, i.e., image classification, dense prediction etc., RF should be sufficiently large to capture the semantic information (highest RF should be at least encompass the raw input). In object detection, RF should be cautiously determined. *We argue that in low-level regression tasks, its unnecessary for the highest RF to cover the whole image but is supposed to be large enough to capture the neighboring for the estimated pixel.* In Fig. 4, we compare the proposed model SRN with the U-Net

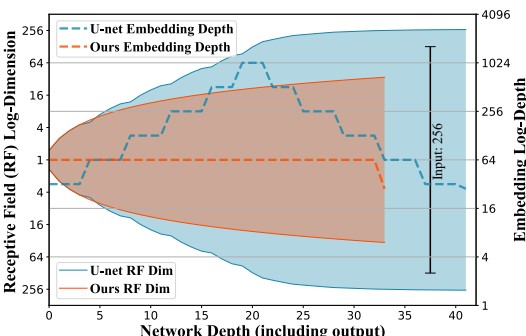

Figure 4: Receptive field (RF) dimension and network embedding depth comparison (U-Net (Miao et al., 2019) v.s. Ours). U-Net unnecessarily observes a super large spatial area, at the end, i.e., 532×532, while SRN finally focuses on a sufficiently large neighborhood for approximation, compared with the input spatial size denoted as a vertical black bar.

utilized by Miao et al. (2019) in terms of the spatial/spectral dimension. The highest RF of the U-Net is much larger than the input (256 v.s. 532), which is oversized and thus improper. The highest RF of proposed SRN, in comparison, makes more sense (256 v.s. 69). In SRN, concurrent with such a linearly expanding behavior of RF, the spatial size intermediate embedding remains the same, which is thus called "Spatial Invariance".

To approximate the 3D signal in a rank minimization manner, previous researchers attempt to reduce the spectral channels somewhere in the model, which turns out to be a lossy compression operation and is avoided in our work. While in the proposed model, we simply treat the data cube as a whole and thus keeps the spectral dimension constant through the main body (spectral invariant) after expanding from $N_\lambda = 28$ to $N_\gamma = 64$.

### 3.2.2 EFFICIENCY MAXIMIZATION: TOWARD PRACTICAL APPLICATIONS

In practical application scenarios, performance and efficiency of the proposed method consist of a paradox for all existing reconstructive methods, including our proposed network. For such a high-performanced (as demonstrated by considerable experiments in 4) and concise network, we seek to further maximize its efficiency at the expense of negligible performance descent. It turns out the metric, floating point operations (FLOPs, *i.e.*, amount of additions and multiplications), provides a favourable perspective of view, which 1) is inversely proportional to the reconstructive rate mentioned in 1.1, 2) is positively related to the computing power. We minimize the FLOPs via introducing rescaling manipulation-pairs, which are consist of a *downsampling* operation, i.e., a convolutional layer with stride 2, and an *upsampling* operation (i.e., a convolutional layer followed by *PixelShuffle* operation (Shi et al., 2016)). Spatial size will be doubled/halved by a single up-sampling/downsampling operation. As annotated by Fig. 3, rescaling manipulations can be flexibly inserted into model, i.e., right after the intermediate embeddings. Except for the initially proposed network SRN v1, we further try two more variants: SRN v2 contains one rescaling pair, which reduces the spatial size of $g_0$ and recovers at $g_\sum$; SRN v3, in which second rescaling pair is inserted in the after $g_4$ and $g_{12}$. Interestingly, experimental results below proves both of the variants are still competitive to state-of-the-art performance (Tab. 1 and Tab. 2) in simulation.

## 4 EXPERIMENTS

**Dataset.** As in Meng et al. (2020b), we adopt the same 28 wavelengths distributed within the range of 450nm to 650nm obtained by spectral interpolation manipulation for the HSIs, and conduct experiments on the following two data type: 1) Simulation Data and 2) Real HSI data.

Table 1: PSNR (dB) values by different algorithms on 10 scenes in the simulation dataset. Our approach is denoted by `v1`: SRN w/o rescaling pair, `v2`: SRN with 1 rescaling pair, and `v3`: SRN with 2 rescaling pair. Notably, `v2` and `v3` are provided to further reduce the FLOPs of `v1` (see Tab. 3) while still exhibiting a competitive performance over seven state-of-the-art methods.

| Method | Scene1 | Scene2 | Scene3 | Scene4 | Scene5 | Scene6 | Scene7 | Scene8 | Scene9 | Scene10 | Avg. |
|---|---|---|---|---|---|---|---|---|---|---|---|
| U-net | 28.28 | 24.06 | 26.02 | 36.33 | 25.51 | 27.97 | 21.15 | 26.83 | 26.13 | 25.07 | 26.80 |
| HSSP | 31.07 | 26.30 | 29.00 | 38.24 | 27.98 | 29.16 | 24.11 | 27.94 | 29.14 | 26.44 | 28.93 |
| $\lambda$-net | 30.82 | 26.30 | 29.42 | 37.37 | 27.84 | 30.69 | 24.20 | 28.86 | 29.32 | 27.66 | 29.25 |
| TSA-Net | 31.26 | 26.88 | 30.03 | 39.90 | 28.89 | 31.30 | 25.16 | 29.69 | 30.03 | 28.32 | 30.24 |
| GSM-based | 32.38 | 27.56 | 29.02 | 36.37 | 28.56 | 32.49 | 25.19 | 31.06 | 29.40 | 30.74 | 30.28 |
| PnP-DIP-HSI | 32.70 | 27.27 | 31.32 | 40.79 | 29.81 | 30.41 | 28.18 | 29.45 | 34.55 | 28.52 | 31.30 |
| GAP-net | 33.62 | 30.08 | 33.07 | 40.94 | 30.77 | 33.60 | 27.41 | 31.25 | 33.56 | 30.36 | 32.47 |
| SRN (`v3`) | 32.85 | 28.61 | 31.27 | 39.42 | 29.93 | 32.81 | 26.26 | 30.87 | 31.74 | 29.84 | 31.36 |
| SRN (`v2`) | 33.16 | 29.08 | 32.19 | 39.81 | 30.43 | 33.22 | 26.69 | 31.50 | 31.69 | 29.98 | 31.77 |
| SRN (`v1`) | **34.42** | **30.98** | **33.11** | **41.58** | **31.87** | **35.38** | **28.26** | **33.38** | **33.75** | **31.75** | **33.45** |

Table 2: SSIM values by different algorithms on 10 scenes in the simulation dataset, which shows the consistency with the above metric PSNR. Both of PSNR and SSIM indicate the promising performance of proposed SRN on syhthetic testing data.

| Method | Scene1 | Scene2 | Scene3 | Scene4 | Scene5 | Scene6 | Scene7 | Scene8 | Scene9 | Scene10 | Avg. |
|---|---|---|---|---|---|---|---|---|---|---|---|
| U-net | 0.822 | 0.777 | 0.857 | 0.877 | 0.795 | 0.794 | 0.799 | 0.796 | 0.804 | 0.710 | 0.803 |
| HSSP | 0.852 | 0.798 | 0.875 | 0.926 | 0.827 | 0.823 | 0.851 | 0.831 | 0.822 | 0.740 | 0.834 |
| $\lambda$-net | 0.880 | 0.846 | 0.916 | 0.962 | 0.866 | 0.886 | 0.875 | 0.880 | 0.902 | 0.843 | 0.886 |
| TSA-Net | 0.887 | 0.855 | 0.921 | 0.964 | 0.878 | 0.895 | 0.887 | 0.887 | 0.903 | 0.848 | 0.893 |
| GSM-based | 0.920 | 0.892 | 0.925 | 0.970 | 0.894 | 0.938 | 0.898 | 0.932 | 0.925 | 0.934 | 0.923 |
| PnP-DIP-HSI | 0.898 | 0.832 | 0.920 | 0.970 | 0.903 | 0.890 | 0.913 | 0.885 | 0.932 | 0.863 | 0.901 |
| GAP-net | 0.926 | **0.914** | **0.944** | 0.966 | 0.925 | 0.936 | **0.915** | 0.918 | 0.937 | 0.914 | 0.929 |
| SRN (`v3`) | 0.906 | 0.850 | 0.903 | 0.954 | 0.911 | 0.927 | 0.839 | 0.913 | 0.909 | 0.893 | 0.900 |
| SRN (`v2`) | 0.909 | 0.852 | 0.902 | 0.943 | 0.911 | 0.927 | 0.838 | 0.918 | 0.910 | 0.893 | 0.900 |
| SRN (`v1`) | **0.931** | 0.906 | 0.923 | **0.970** | **0.941** | **0.958** | 0.867 | **0.953** | **0.937** | **0.939** | **0.932** |

For simulation data, both CAVE (Park et al., 2007) synthetic dataset and KAIST (Choi et al., 2017) synthetic dataset are applied in our simulation experiment. Regarding the training set, we create 205 1024×1024×28 large image examplers from 30 256×256×28 images from CAVE dataset by randomly concatenating. Operations like rotation and rescaling are both used to magnify the robustness. Training samples of size 256×256×28 will be further randomly cropped from the examplers. We compare with other methods on the same ten different 256×256×28 HSIs abstracted from KAIST dataset. For real HSI data, we augment the above training set by additional 37 HSI images in KAIST (Choi et al., 2017) dataset, all of which will be cropped into 660×660×28 to match the real-world measurement, which are abstracted by CASSI system developed in Meng et al. (2020b).

**Training and Testing Procedure.** The model is trained to minimize a mean squared error (MSE) between the ground truth and output. For simulation experiment, reconstructive inputs can be computationally acquired by mimicking the actual CASSI reproduced in Meng et al. (2020b). Notably, the real-world captured measurements are disturbed by the noise introduced by the optical system. Therefore, we add Gaussian noise with standard deviation randomly generated from the range $[0, 0.05]$ to mimic this scenario. For both simulation training/testing and real-wrold testing, the identical mask are employed for a fair comparison.

**Compared Methods**. We compare with seven state-of-the-art reconstruction algorithms, including U-Net (Ronneberger et al., 2015), HSSP (Wang et al., 2019), $\lambda$-net (Miao et al., 2019), TSA-Net (Meng et al., 2020b), GSM-based method (Huang et al., 2021), PnP-DIP-HSI (Meng et al., 2021) and GAP-net (Meng et al., 2020a), among which $\lambda$-net, TSA-Net and GSM-based methods yield plausible results. Peak Signal-to-Noise Racial (PSNR) and Structural SIMilarity (SSIM) (Wang et al., 2004) are used for a quantitative comparison. The PSNR is computed by

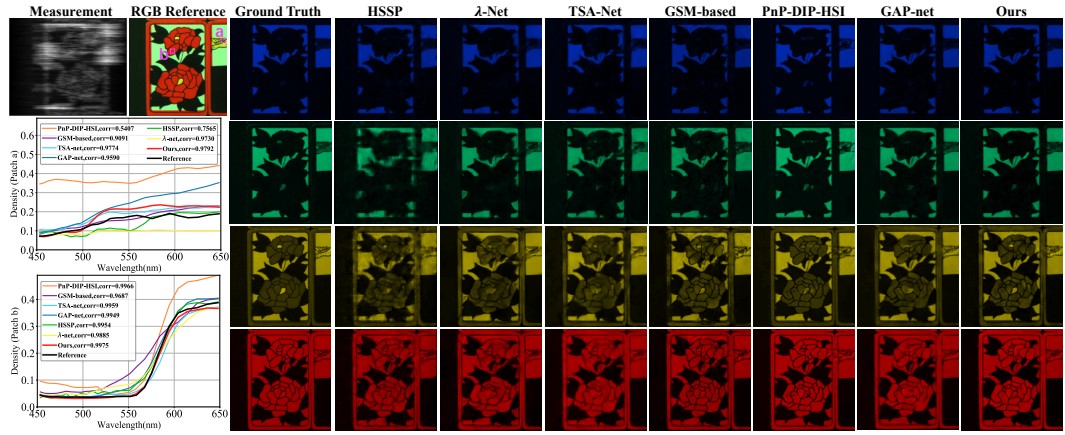

Figure 5: Comparison of reconstructed results for synthetic HSI on the *flower pating* scene. Six state-of-the-art methods and our proposed method are included. The RGB reference is shown to demonstrate the color (top-left). Our results recover the most details contained in real scene. For the chosen regions $a$ (yellow patch) and $b$ (red patch), density curves of our result (bottom-left) show highest correlations with the references (0.9792, 0.9975), which indicates the effectiveness of our method within corresponding waveband (yellow: 565nm∼590nm; red: 625nm∼740nm).

$PSNR_{ch} = 10\log_{10}(\frac{MAX_I^2}{MSE_{ch}})$, where the channel-wise PSNR value will be gathered for average computing and $MAX_I^2$ denotes the maximum pixel value in ground truth image $I$.

We implemented our model by PyTorch and trained it with the ADAM optimizer. There are $\boldsymbol{K}$=16 network embeddings in main body. The learning rate was initialized as $4\times10^{-4}$, decreased by half every 50 epochs, for both simulation/real settings. We set batch size = 4 for the best validation performance. The training is conducted on a NVIDIA TIATN RTX GPU on simulation data.

## 4.1 EXPERIMENTS ON SYNTHETIC DATA

As shown in both Tab. 1 and 2, the proposed method (*i.e.*, our best model without rescaling, denoted as SRN `v1`) has achieved the highest averaged PSNR and SSIM of 10 synthetic testing samples as 33.45 dB and 0.932, respectively, which is an considerable breakthrough. Although the other two versions of our model (*i.e.,* the proposed method with 1 or 2 rescaling pairs, termed as, SRN `v2` and SRN `v3`) are mainly designed for reducing the computational cost, they also exhibit the competitive performance compared with existing methods. Validated by three variants of our model, the proposed method demonstrated great stability and effectiveness.

Fig. 5 visualizes the reconstructed results between six compared methods and our model. Our reconstructed images contain the most details among four chosen spectral channels including 462.1nm, 498nm, 575.3nm and 625.1nm. We also plot the spectral density curves corresponding to two picked regions ( box "a" and "b" in Fig. 5). The highest correlations (*i.e.*, 0.9792, 0.9975) and highly coincidence between our curves and reference proves the spectral-wise effectiveness of our model. To clearly check the reconstructive ability of the proposed model (i.e., (`v1`)), we put RGB reconstructions of eight 256×256×28 reconstructed simulation results in Fig. 7.

## 4.2 EXPERIMENTS ON REAL DATA

We validate the effectiveness of proposed method on hyperspectral images collected from real scenes. We show one example of *legoman* in Fig. 6 (a) by comparing our method with three best reconstructive methods, $\lambda$-net (Miao et al., 2019), TSA-Net (Meng et al., 2020b) and GSM-based method (Huang et al., 2021), on 8 spectral channels. Our reconstructive results are perceptually complete and demonstrate few artifact, which is obvious on the "face" of the legoman. We pick a small red region for spectral-wise performance verification and gets the highest correlation with the reference generated by a spectrometer.

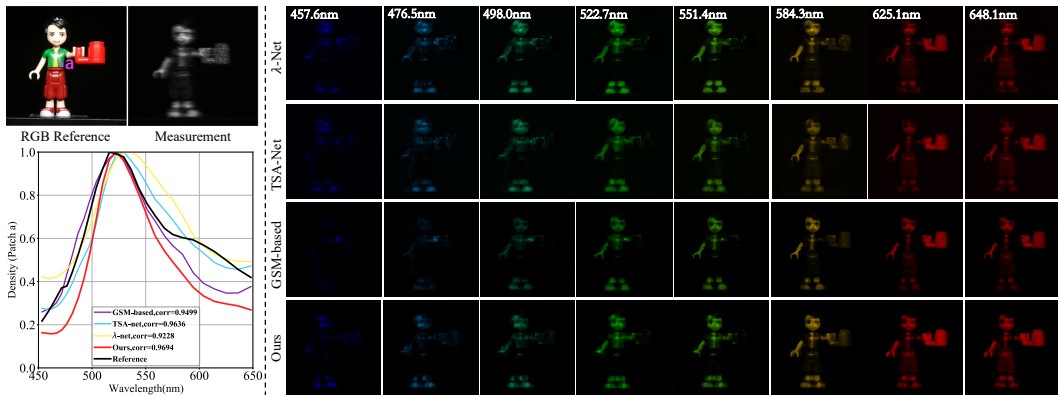

Figure 6: Real reconstructed hyperspectral images comparison between $\lambda$-net, TSA-Net, GSM-based method and our method from the real-captured CASSI measurement of *legoman* scene. The RGB reference is shown to demonstrate the color. By comparison, our results give little distortion and artifact. Within the chosen region $a$ (green patch), our reconstructed pixels yield the highest correlation with the reference given by spectrometer, indicating the effectiveness of our method within waveband of green, i.e., 500nm∼565nm.

### 4.3 EFFICIENCY ANALYSIS

**Model size.** Besides the breakthroughs in terms of performance, we further compare the model size and floating-point operations (FLOPs) between four E2E deep-learning-based methods and three variants of our proposed method. As shown in Tab. 3, the proposed SRN v1 (*i.e.*, the one without rescaling pairs) contains only 2.82% trainable parameters compared with TSA-Net, less than 1/3 params as GSM-based method. We reduces 96% and 98% trainable parameters when compared with U-net (Ronneberger et al., 2015) and $\lambda$-net, respectively. The other two variants involve a little bit more trainable parameters due to the additional covolutional layers introduced by the rescaling pairs. Our models requires much less computing power and thus can be flexibly deployed, which benefits the real-time reconstruction. We visualize the reconstructed high-resolution HSI, i.e.,1024×1024×28 in A.4.

| Method | #params (M) | FLOPs (G) | PSNR (dB) |
|---|---|---|---|
| U-net | 31.32 | 58.99 | 26.80 |
| $\lambda$-net | 62.64 | 117.98 | 29.25 |
| TSA-Net | 44.25 | 80.08 | 30.24 |
| GSM-based | 3.76 | 646.35 | 30.28 |
| SRN v1 | 1.25 | 81.84 | 33.45 |
| SRN v2 | 1.44 | 25.07 | 31.77 |
| SRN v3 | 1.62 | 18.57 | 31.36 |

Table 3: Analysis efficiency of the deep learning-based algorithms, with a spatial size of input as 256×256. The annotation v1: SRN w/o rescaling pair, v2: SRN with 1 rescaling pair, and v3: SRN with 2 rescaling pair.

**Floating-Point Operations (FLOPs).** Given the input of spatial shape 256×256, we count the total amount of addition and multiplication for different deep learning-based model. As shown in Tab. 3, the FLOPs of the proposed model without rescaling pairs is 81.84G, which are slightly more than that of TSA-Net. The underlying intuition is that though the proposed model is rather small, the amount of operations conducted in the model is still formidable, which limits the capacity of processing large image patches. Similar problem shows on GSM-based method, which contains over 30× FLOPs than SRN v3. As shown in Tab. 3, two variants (*i.e.*, the bottom two models, SRN v2 and SRN v3) mitigate this issue successfully. Typically, we reduce 69.4% FLOPs by using one rescaling pairs, which is less than 1/3 the FLOPs of TSA-Net. If we further add one more rescaling pair, the resulting FLOPs will be only 18.57G.

## 5 CONCLUSION AND FUTURE WORKS

We provided a simple yet promising method specially-designed for HSI reconstruction in this paper. The proposed method outperformed existing methods, by setting a new state-of-the-art performance with a negligible amount of model parameters. We alleviated the problem of inefficiency of neural network-based reconstructive methods and thus make a step forward to practical application of HSI reconstruction. Our method can not only be solely referenced as a baseline, also conjointly served as a qualified backbone for future research in the HSI community.

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

Table 4: Averaged PSNR/dB (↑), SSIM (↑), model size/M (↓) and FLOPs/G (↓) on *Simplified Res-Block* (without Batch Bormalization) and normal ResBlock (containing Batch Normalization) in simulation dataset. Batch size is kept as 4, learning rates is initialized as 0.0004 and will be reduced by half every 50 epochs throughout the experiment. Apparently, there is no benefit for adding Batch Normalization layer.

| Batch Normalization | SRN v1 | | | | SRN v2 | | | | SRN v3 | | | |
|---|---|---|---|---|---|---|---|---|---|---|---|---|
| | PSNR | SSIM | # params | FLOPs | PSNR | SSIM | # params | FLOPs | PSNR | SSIM | # params | FLOPs |
| SRN w/ BN | 32.40 | 0.910 | 1.255 | 81.839262 | 31.69 | 0.894 | 1.440 | 25.065163 | 29.65 | 0.870 | 1.624 | 18.572380 |
| SRN w/o BN | **33.45** | **0.932** | **1.251** | **81.839260** | **31.77** | **0.900** | **1.436** | **25.065161** | **31.36** | **0.900** | **1.620** | **18.572378** |

# A    APPENDIX

## A.1    EXPERIMENTAL SETTINGS

**CAVE dataset.** We used CAVE dataset (Park et al., 2007) to train our model for both simulation and real data experiments. CAVE dataset contains 30 512×512 multispectral images corresponding to wavelength of 400nm∼700nm. We generate 175 examplers of 1024×1024 spatial size by random concatenation. To increase the robustness of the training set, images are randomly rotated before concatenation. Besides, another 30 examplers are generated via directly rescaling the original ones. For simulation, image patches of 256×256 spatial size are randomly cropped and then fed into the model. For real hyperspectral images reconstruction, 660×660 images patches are randomly cropped to be the training samples of our CNN model. For high-resolution image reconstruction, 1024×1024×28 examplers are directly fed into the model as training samples. Fig. 8 (a) shows a rescaled training exampler at the wavelength of 453.3nm and (b) shows a concatenated training exampler at the wavelength of 522.7nm.

**KAIST dataset.** KAIST (Choi et al., 2017) dataset contains 30 high-quality hyperspectral images of spatial size 2704×3376. In simulation experiment, ten image patches of 256×256×28 are used as the testing data. For real HSI reconstruction, 36 image patches of spatial size 1024×1024 are abstracted and combined with above 205 sample for training purpose. For high-resolution image reconstruction, 17 samples of spatial size 1024×1024 are used as the testing data. Fig. 8 (c) demonstrates an example of testing image patch at 498.0nm wavelength, which is used for the third task.

**Evaluation metric.** Two widely-used validation criteria, Peak Signal-to-Noise Ratio (PSNR) and Structural Similarity (SSIM) are used for quantitative comparison. SSIM is computed by

$$\text{SSIM}(x, y) = \left[ \frac{2\mu_x \mu_y + C_1}{\mu_x^2 + \mu_y^2 + C_1} \right]^\alpha$$
$$\cdot \left[ \frac{2\sigma_x \sigma_y + C_2}{\sigma_x^2 + \sigma_y^2 + C_2} \right]^\beta \cdot \left[ \frac{\sigma_{xy} + C_3}{\sigma_x \sigma_y + C_3} \right]^\gamma, \tag{5}$$

where $\mu_x$, $\mu_y$ are local means, $\sigma_x$, $\sigma_y$ are standard deviations, $\sigma_{xy}$ is cross-covariance for image $x$ and $y$.

## A.2    HYPER-PARAMETER TUNING

In this experiment, we perform a number of trials on simulation dataset, to find the optimal hyperparameter setting for the proposed reconstructive models. Specifically, ablation studies on batch normalization layer 4, initial learning rate of 0.0002, 0.0004, 0.0006 (see Table 5 for averaged PSNR and SSIM values), and batch size of 2,4,8 (see Table 6 for averaged PSNR and SSIM values) are involved in three versions of proposed model (SRN v1, SRN v2, SRN v3).

Besides, to further validate the superiority of *Simplified ResBlcok (RB)* over normal ResBlock, we compare the

| learning rate | SRN v1 | | SRN v2 | | SRN v3 | |
|---|---|---|---|---|---|---|
| | PSNR | SSIM | PSNR | SSIM | PSNR | SSIM |
| 0.0002 | 32.88 | 0.924 | **32.26** | **0.913** | **31.75** | **0.906** |
| 0.0004 | **33.45** | **0.932** | 31.77 | 0.900 | 31.36 | 0.900 |
| 0.0006 | 32.02 | 0.907 | 32.26 | 0.913 | 30.08 | 0.870 |

Table 5: Averaged PSNR (dB) and SSIM values for different learning rate in simulation experiment, where batch size is kept as 4 and the learning rates is reduced by half every 50 epochs.

Table 7: PSNR (dB) and SSIM values on 1024×1024×28 simulation testing data. Reconstruction is conducted by SRN `v2` model (*i.e.*, the model with one rescaling pair), where optimal hyperparameter settings argued in Section A.2 are leveraged in the training phase.

| scene | 1 | 2 | 3 | 4 | 5 | 6 | 7 | 8 | 9 | 10 | 11 | 12 | 13 | 14 | 15 | 16 | 17 | *Avg.* |
|---|---|---|---|---|---|---|---|---|---|---|---|---|---|---|---|---|---|---|
| PSNR | 28.78 | 29.97 | 30.40 | 31.74 | 33.31 | 30.96 | 34.56 | 34.46 | 34.49 | 27.28 | 34.95 | 33.07 | 31.88 | 33.36 | 29.20 | 29.85 | 30.03 | 31.66 |
| SSIM | 0.869 | 0.913 | 0.943 | 0.937 | 0.956 | 0.958 | 0.958 | 0.962 | 0.956 | 0.821 | 0.945 | 0.948 | 0.819 | 0.928 | 0.937 | 0.886 | 0.926 | 0.921 |

model size, FLOPs (FLoating-point OPerations) and quantitative evaluations (averaged PSNR and SSIM) on both settings by only adding the Batch Normalization (Ioffe & Szegedy, 2015) layer behind the first convolutional layer in each *Simplified ResBlock (RB)* (shown in Table 4). For comparison, TSA-Net (Meng et al., 2020b) achieves 30.15dB and 0.893 respectively, with 44.25M parameters and 80.08G FLOPs. Obviously, Batch Normalization layers not only result in more parameters and FLOPs, also undermine the performance. As demonstrated in the three tables above, we choose the optimal hyperparameter setting according to our best model's performance.

$$\text{learning rate} = 0.0004,$$
$$\text{batch size} = 4. \tag{6}$$

## A.3 STATE-OF-THE-ART PERFORMANCE

In this section, we visualize more reconstructed HSIs produced by our best model, which is SRN `v1` (*i.e.*, the original model without rescaling pair). Firstly, we add one additional simulation and one additional real reconstructed results with spectral analysis (i.e., density curves) in Fig. 9 and Fig. 10. For comparison, the same sixteen spectral channels of reconstructed HSIs generated by λ-net (Miao et al., 2019), TSA-Net (Meng et al., 2020b) and GSM-based method (Huang et al., 2021) are also plotted, shown in Fig. 11~Fig. 20. Better visualization performance can be achieved by zooming in.

| batch size | SRN `v1` | | SRN `v2` | | SRN `v3` | |
|---|---|---|---|---|---|---|
| | PSNR | SSIM | PSNR | SSIM | PSNR | SSIM |
| 2 | 32.24 | 0.909 | 31.44 | 0.894 | 31.02 | 0.888 |
| 4 | **33.45** | **0.932** | **31.77** | 0.900 | 31.36 | 0.900 |
| 8 | 32.05 | 0.909 | 31.68 | **0.902** | **31.51** | **0.905** |

Table 6: Averaged PSNR (dB) and SSIM values on different batch sizes in simulation dataset. Initial learning rate is kept as 0.0004 and reduced by half every 50 epochs.

## A.4 HIGH-RESOLUTION HSI RECONSTRUCTION

As compared in Fig. 1 *right*, our model outperforms the TSA-Net (Meng et al., 2020b) on simulation data of spatial size 256×256 with only 2.82% trainable parameters. As compared in Tab. **??**, we outperforms GSM and concretely reduce FLOPs by $> 34 times$, by comparison. Besides, the variants of original model, which are equipped with rescaling pairs, can still achieve promising performance with less than 1/3 FLOPs, as shown in Tab. 1 and Tab. 2. Both of the minimal model size and insignificant amount of computing enables our method a perfect candidate for higher-resolution hyperspectral image reconstruction. Therefore, we setup a simulation experiment—by directly using previous 1024×1024×28 examplers as training samples, we reconstruct high-resolution HSI accordingly through SRN `v2` model (*i.e.*, the model with one rescaling pair). The testing set is composed of 17 images patches of size 1024×1024×28 abstracted from KAIST dataset. We also generate a 1024×1024 noise map as mask for training and testing. As far as we know, this is the first time to reconstruct HSI at such a high resolution using deep learning-based method. Table 7 reports the PSNR and SSIM values on all 17 scenes. Fig. 21~Fig. 26 show the reconstructed images on 16 selected spectral channels.

Table 8: PSNR (dB) and SSIM values on 256×256×28 simulation testing set. To create a well-trained model that can easily adapt to diverse masks, We train SRN v2 model with masks generated from the same distribution, during which optimal hyperparameter settings (*i.e.*, learning rate is 0.0004, batch size is 4) argued in Section A.2 are leveraged.

| scene | 1 | 2 | 3 | 4 | 5 | 6 | 7 | 8 | 9 | 10 | *Avg.* |
|-------|-------|-------|-------|-------|-------|-------|-------|-------|-------|-------|-------|
| PSNR | 32.56 | 28.41 | 31.18 | 39.19 | 29.51 | 33.21 | 26.32 | 31.27 | 32.05 | 29.82 | 31.35 |
| SSIM | 0.901 | 0.841 | 0.899 | 0.948 | 0.898 | 0.927 | 0.837 | 0.918 | 0.908 | 0.893 | 0.897 |

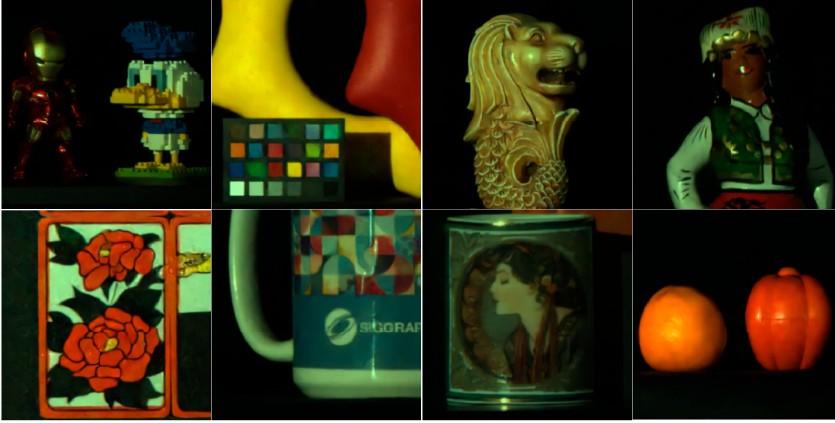

Figure 7: RGB reconstruction of eight reconstructed simulation results.

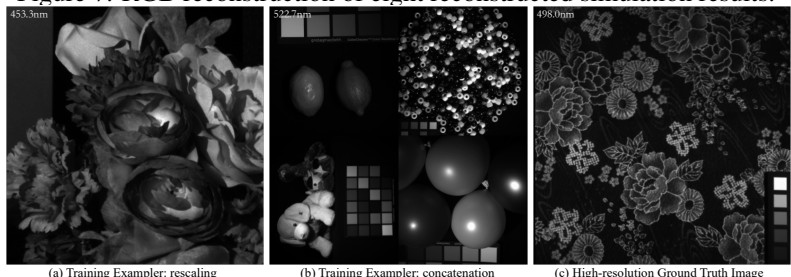

Figure 8: 1024×1024 training/testing hyperspectral images for different tasks, shown on specific wavelengths.

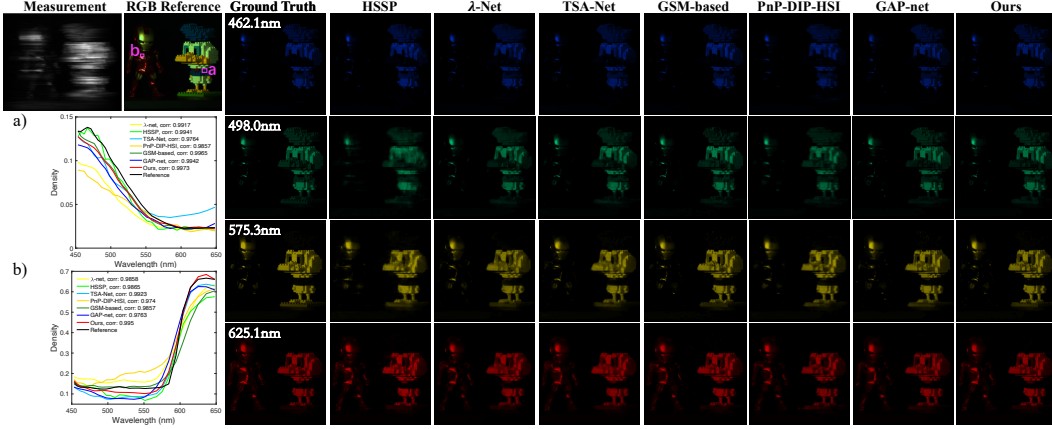

Figure 9: Comparison of reconstructed results for synthetic HSI (*Ironman and Donald Duck* scene). Six state-of-the-art methods and our method (right-column) are included. The RGB reference is shown to demonstrate the color (Top-left). The spectral curves (bottom-left) are corresponding to the boxes denoted in measurement.

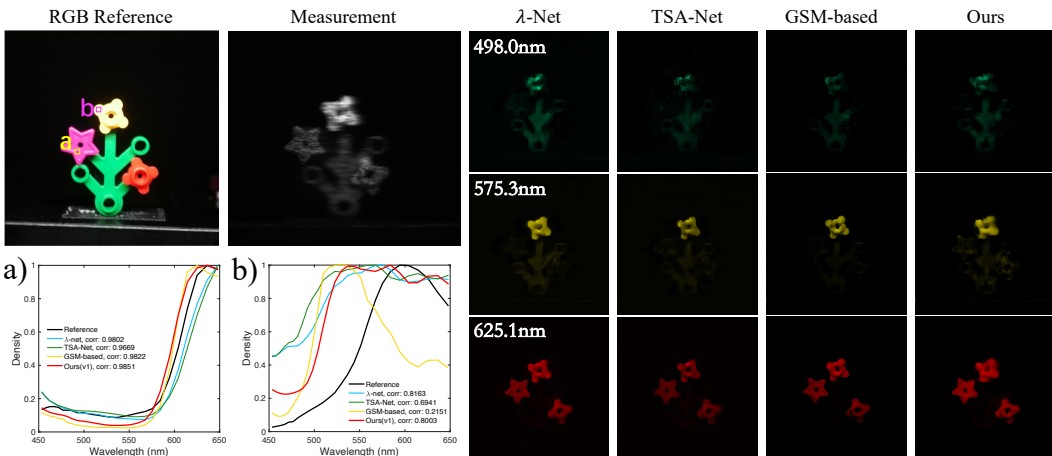

Figure 10: Reconstructed real HSI comparison between λ-net, TSA-Net, GSM-based method and our proposed model v1, (*i.e.*, with one rescaling pair) from the real CASSI measurement (top-left).

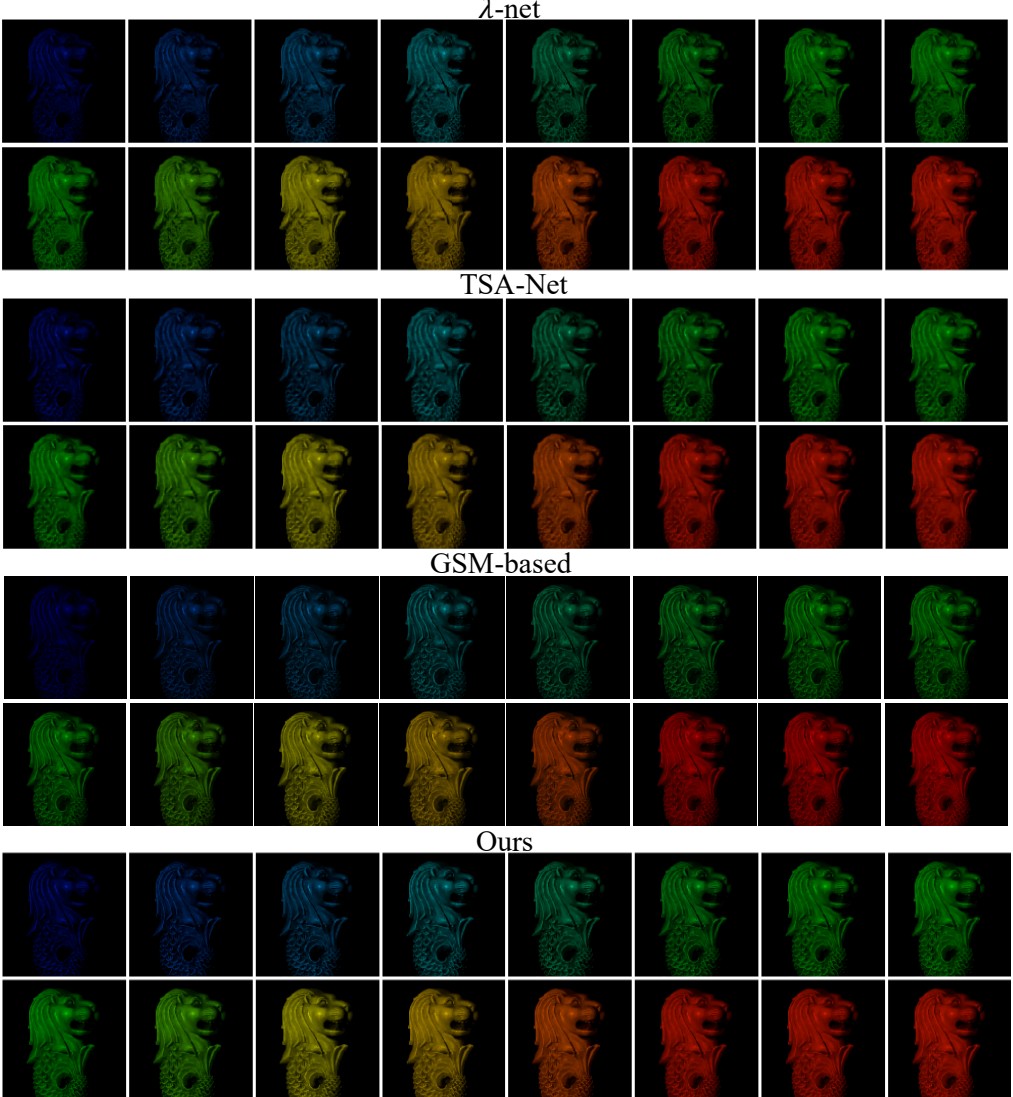

Figure 11: Comparison of reconstructed results produced by $\lambda$-net Miao et al. (2019), TSA-Net Meng et al. (2020b),GSM-based method (Huang et al., 2021) and our best model on simulation dataset scene 1. Better visualization performance can be achieved by zooming in.

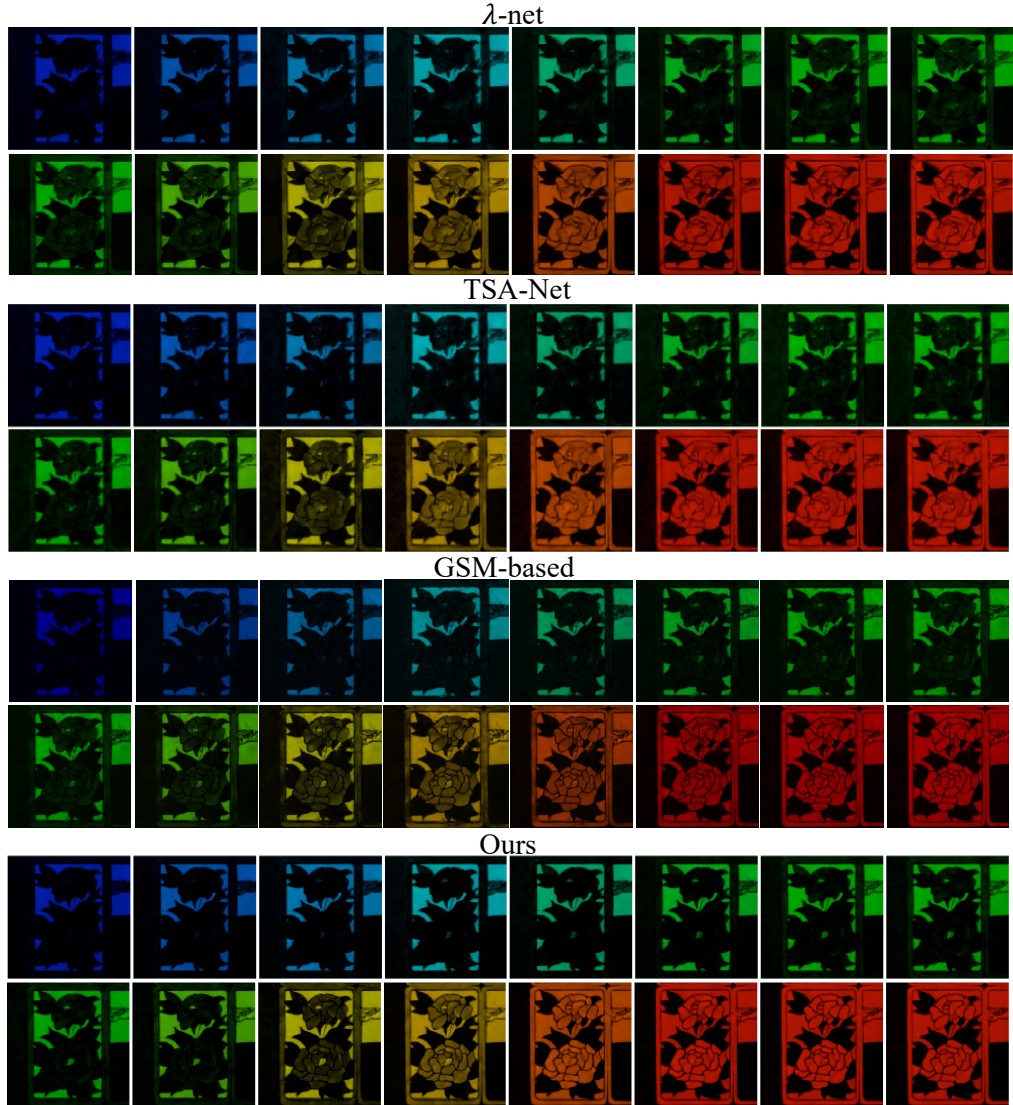

Figure 12: Comparison of reconstructed results produced by $\lambda$-net Miao et al. (2019), TSA-Net Meng et al. (2020b),GSM-based method (Huang et al., 2021) and our best model on simulation dataset scene 2. Better visualization performance can be achieved by zooming in.

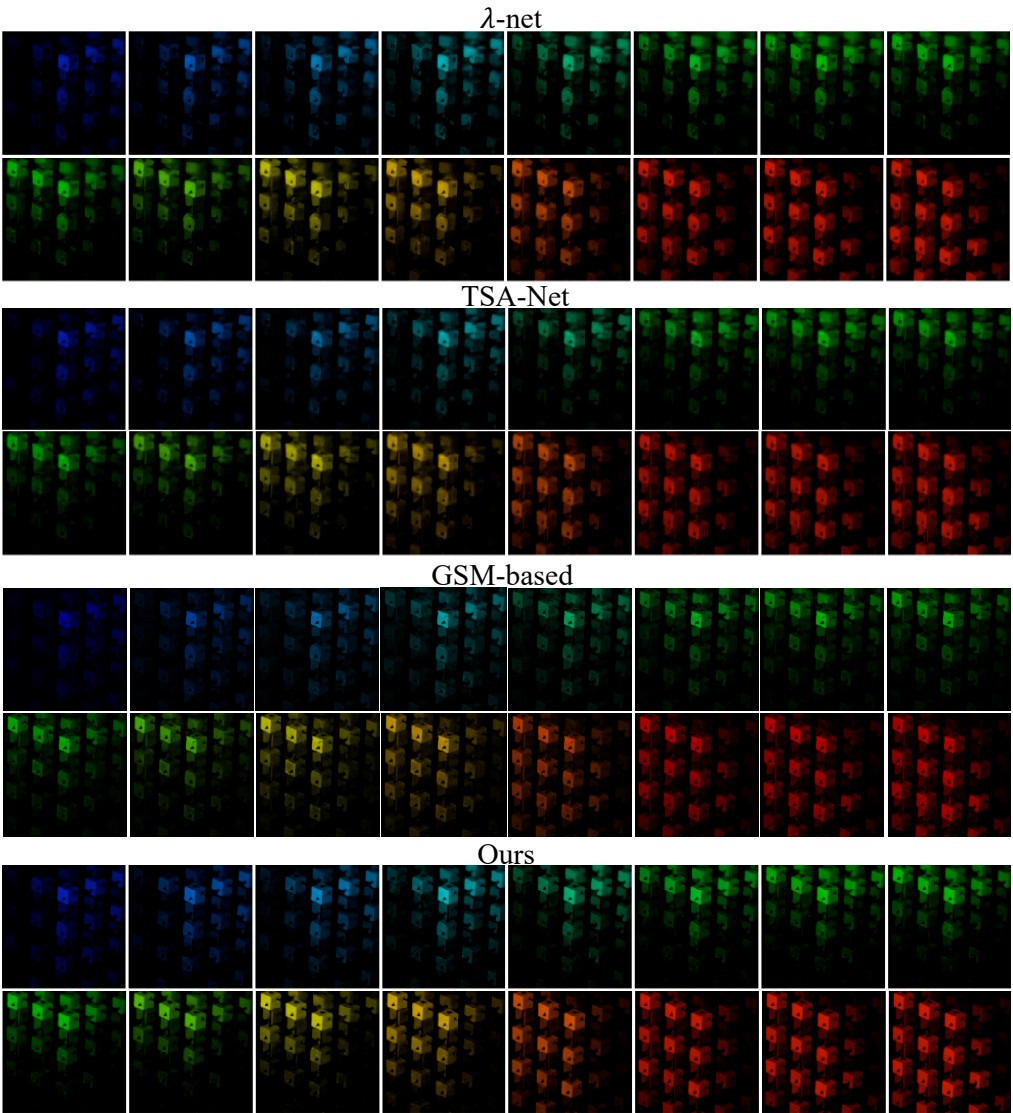

Figure 13: Comparison of reconstructed results produced by λ-net Miao et al. (2019), TSA-Net Meng et al. (2020b),GSM-based method (Huang et al., 2021) and our best model on simulation dataset scene 3. Better visualization performance can be achieved by zooming in.

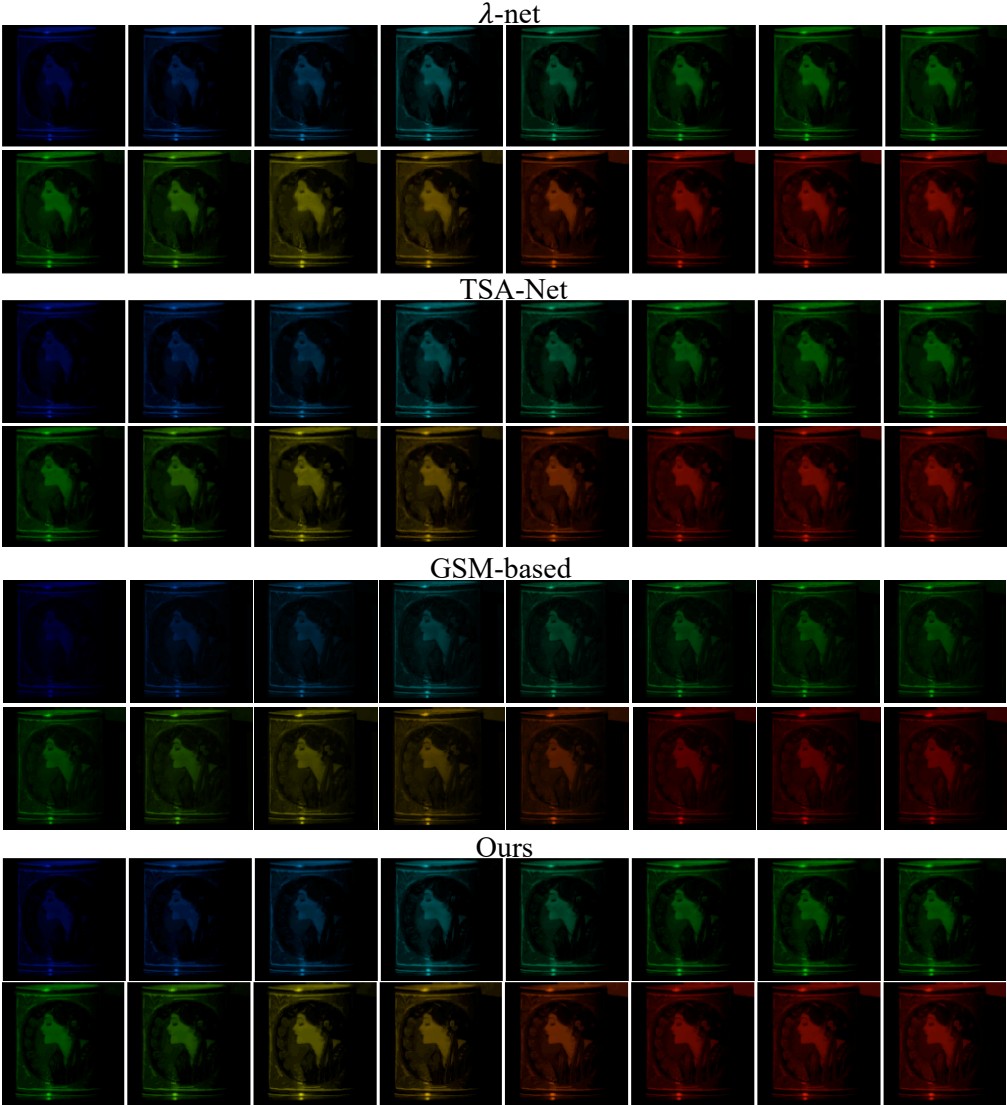

Figure 14: Comparison of reconstructed results produced by $\lambda$-net Miao et al. (2019), TSA-Net Meng et al. (2020b),GSM-based method (Huang et al., 2021) and our best model on simulation dataset scene 4. Better visualization performance can be achieved by zooming in.

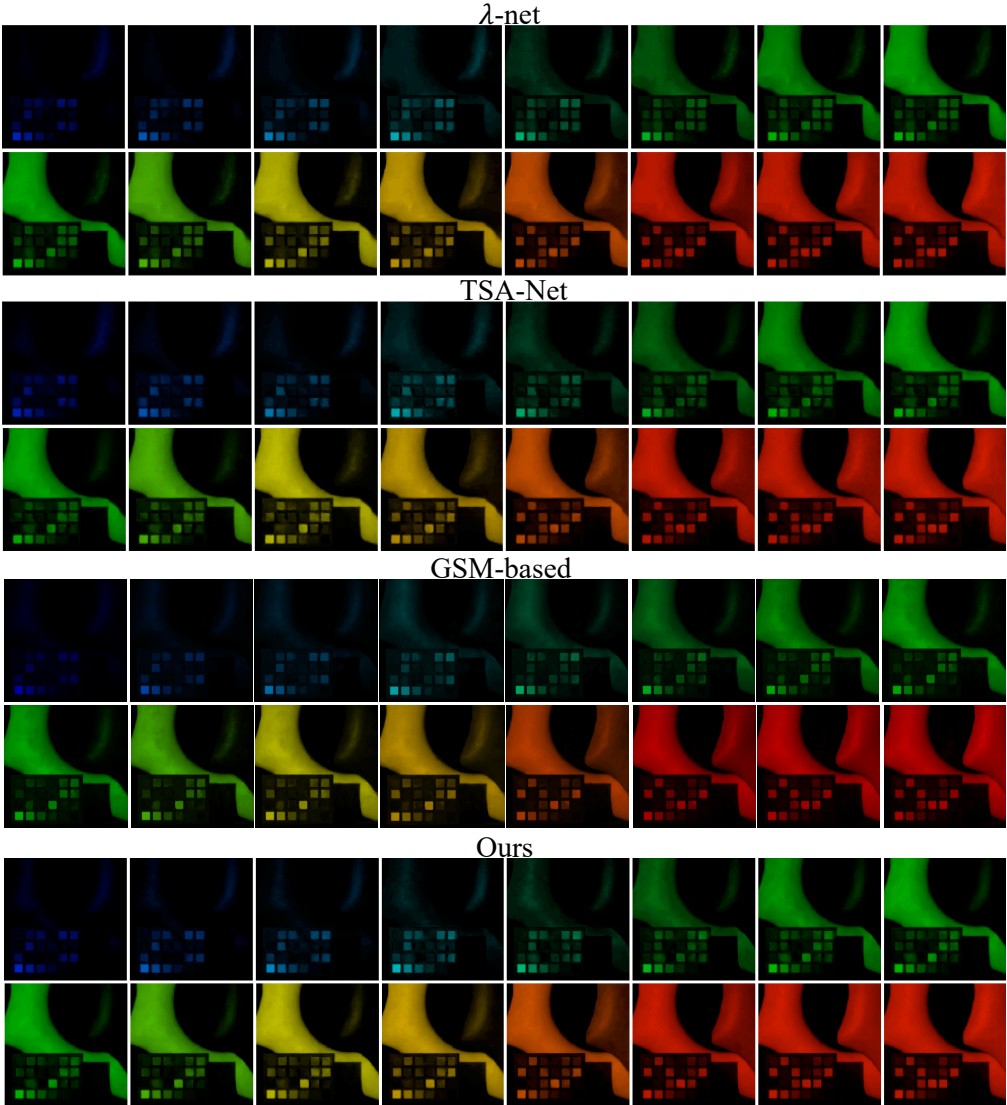

Figure 15: Comparison of reconstructed results produced by λ-net Miao et al. (2019), TSA-Net Meng et al. (2020b),GSM-based method (Huang et al., 2021) and our best model on simulation dataset scene 5. Better visualization performance can be achieved by zooming in.

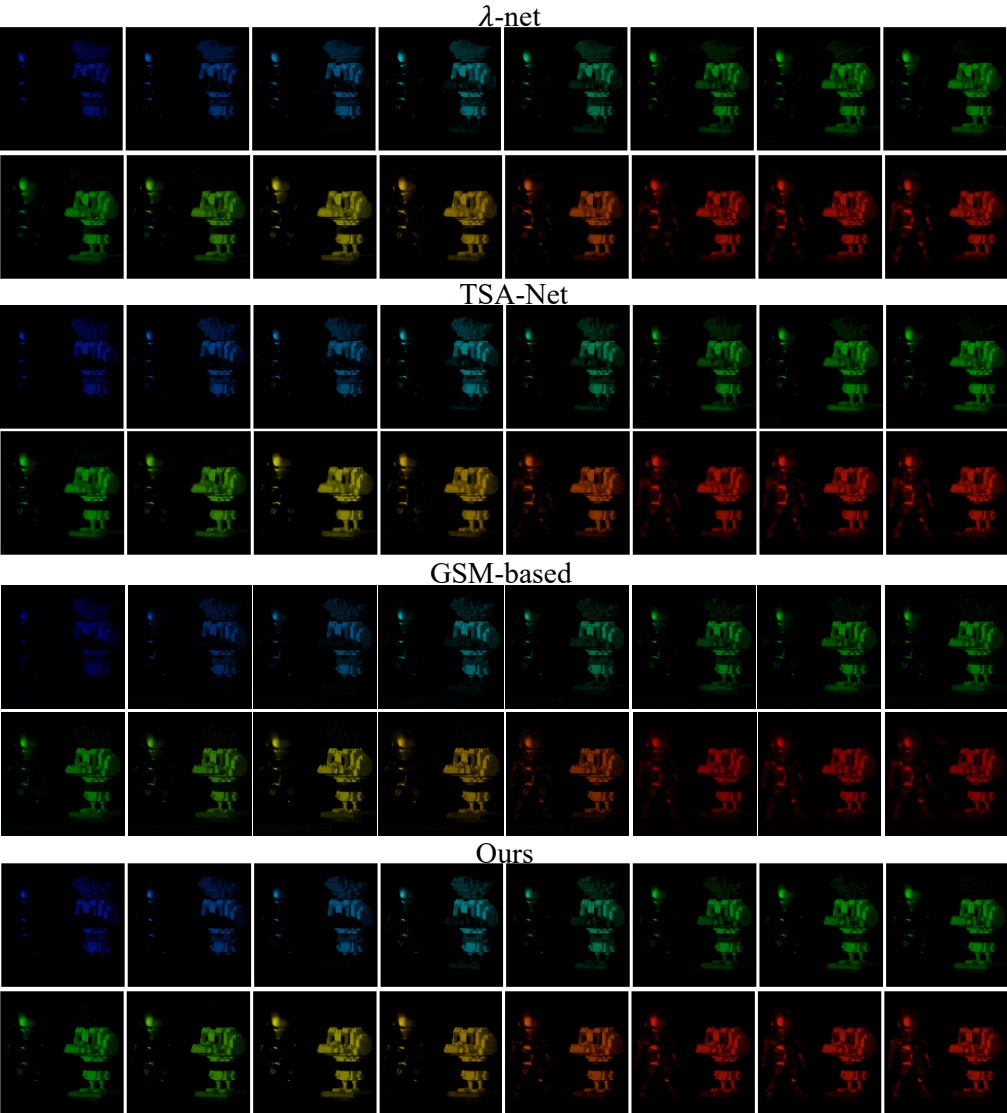

Figure 16: Comparison of reconstructed results produced by λ-net Miao et al. (2019), TSA-Net Meng et al. (2020b),GSM-based method (Huang et al., 2021) and our best model on simulation dataset scene 6. Better visualization performance can be achieved by zooming in.

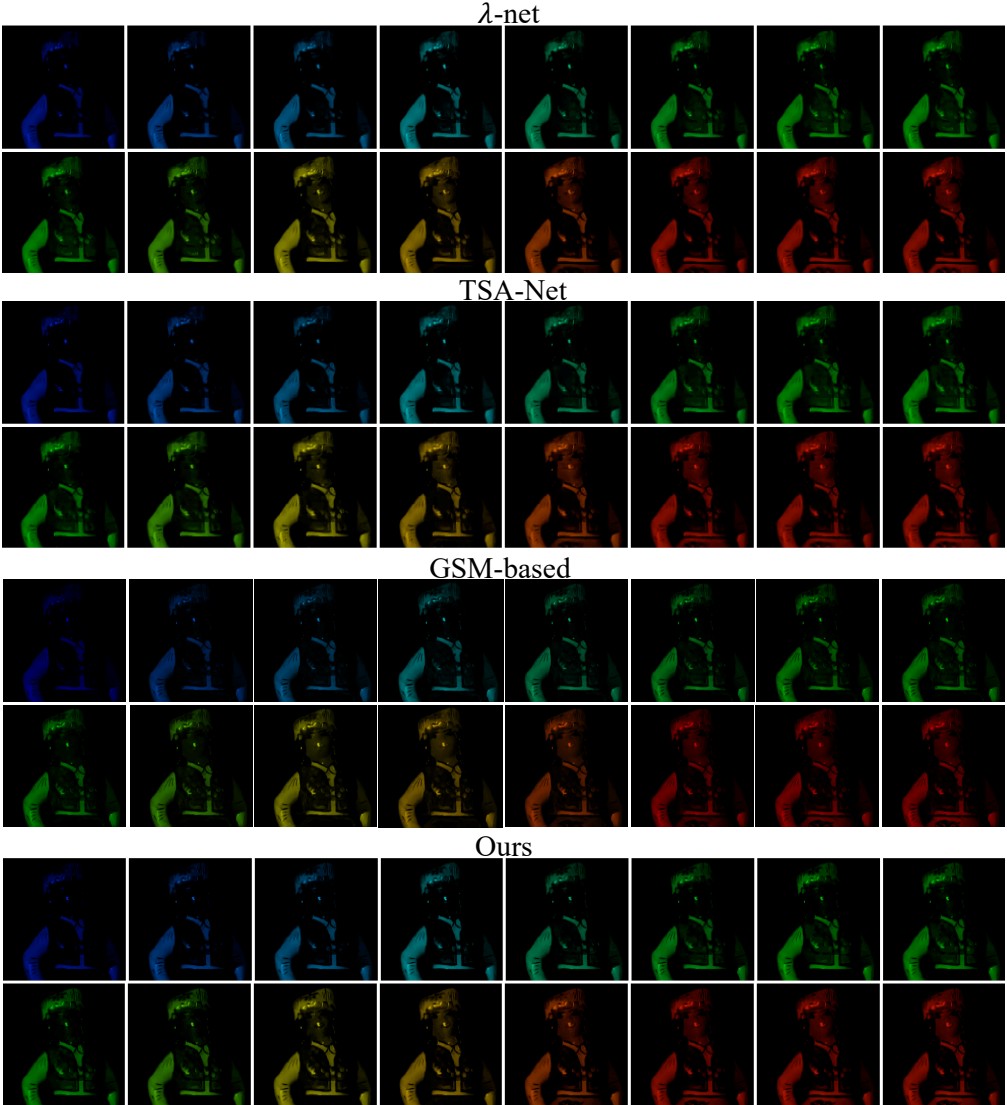

Figure 17: Comparison of reconstructed results produced by $\lambda$-net Miao et al. (2019), TSA-Net Meng et al. (2020b),GSM-based method (Huang et al., 2021) and our best model on simulation dataset scene 7. Better visualization performance can be achieved by zooming in.

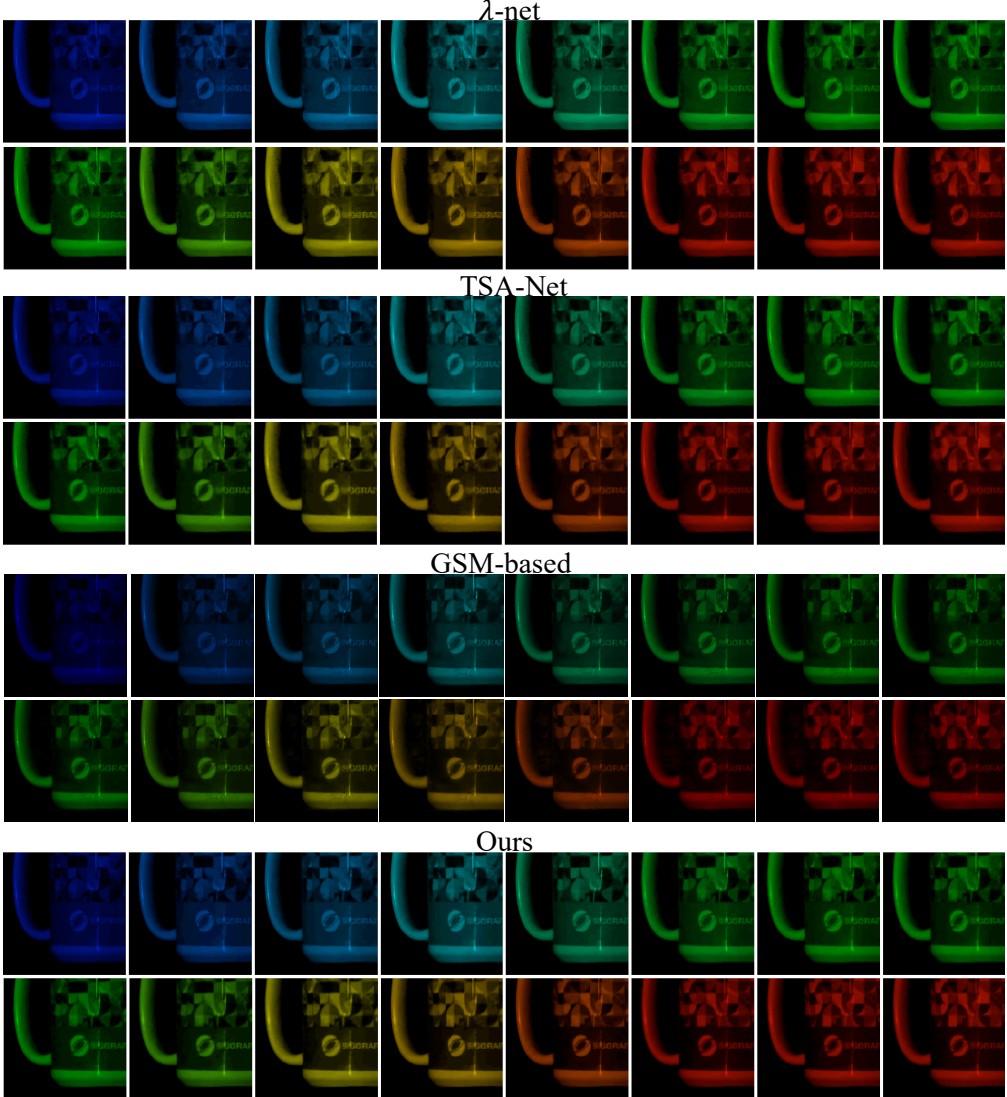

Figure 18: Comparison of reconstructed results produced by $\lambda$-net Miao et al. (2019), TSA-Net Meng et al. (2020b),GSM-based method (Huang et al., 2021) and our best model on simulation dataset scene 8. Better visualization performance can be achieved by zooming in.

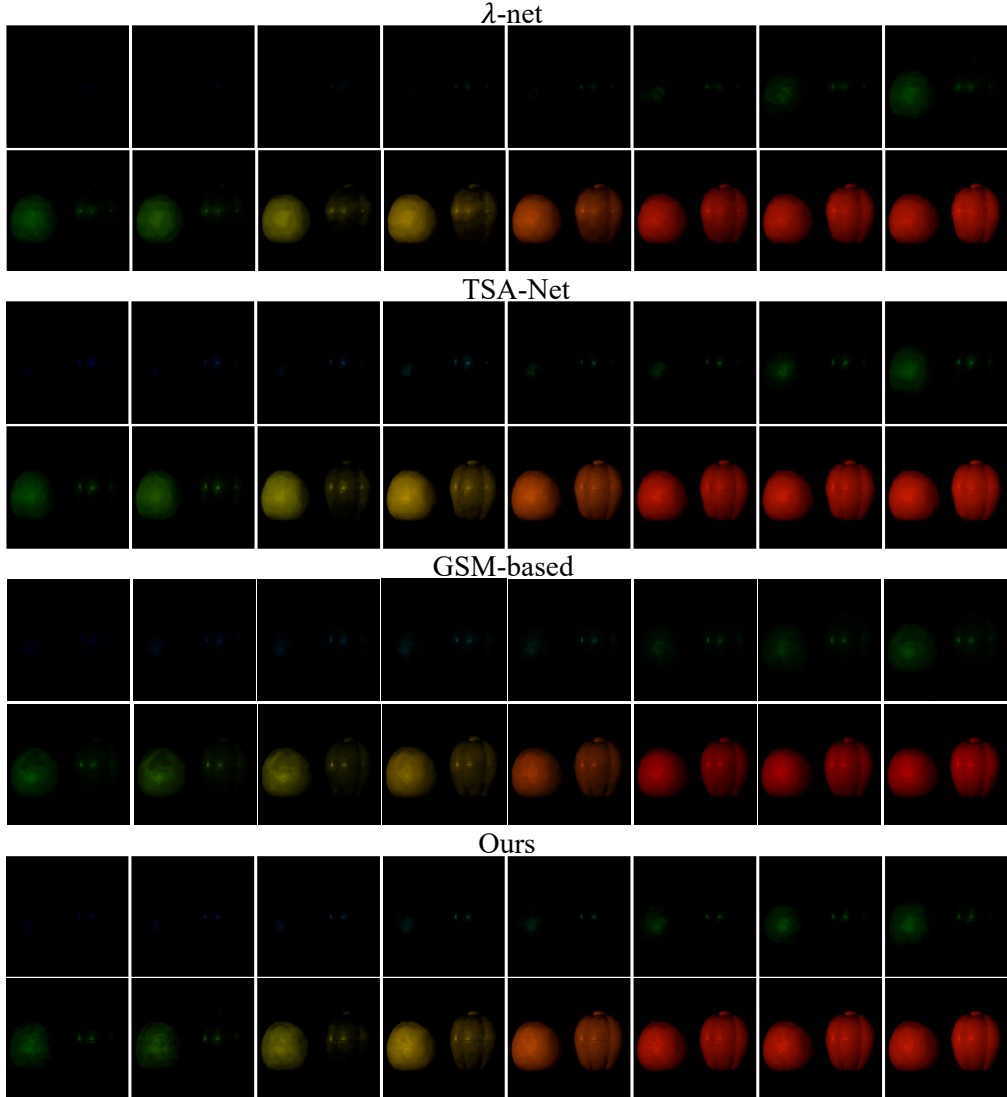

Figure 19: Comparison of reconstructed results produced by $\lambda$-net Miao et al. (2019), TSA-Net Meng et al. (2020b),GSM-based method (Huang et al., 2021) and our best model on simulation dataset scene 9. Better visualization performance can be achieved by zooming in.

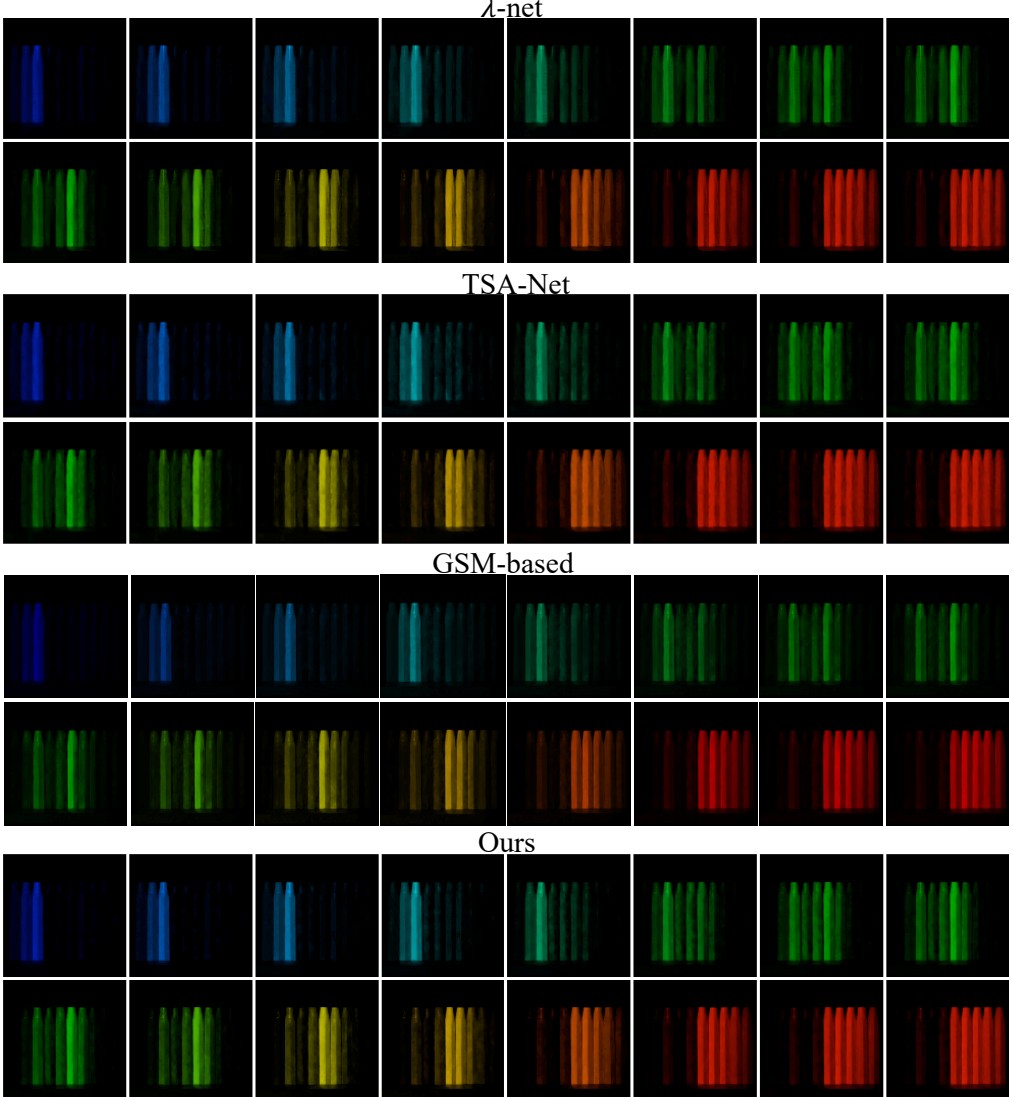

Figure 20: Comparison of reconstructed results produced by $\lambda$-net Miao et al. (2019), TSA-Net Meng et al. (2020b),GSM-based method (Huang et al., 2021) and our best model on simulation dataset scene 10. Better visualization performance can be achieved by zooming in.

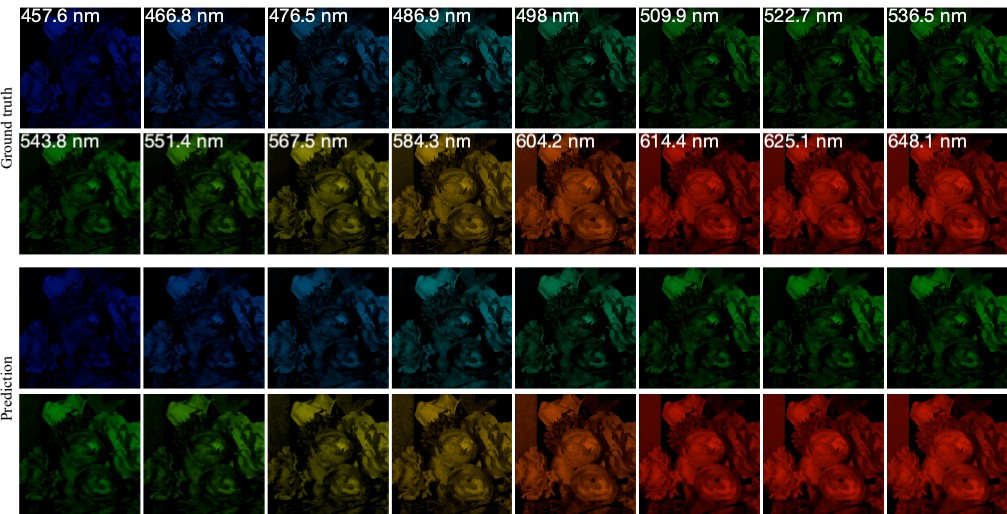

Figure 21: 1024×1024×28 high-resolution HSI reconstruction (scene 4): Sixteen spectral channels of ground truth HSI (upper two rows) and corresponding reconstructive results by SRN v2 model (lower two rows). Better visualization performance by zooming in.

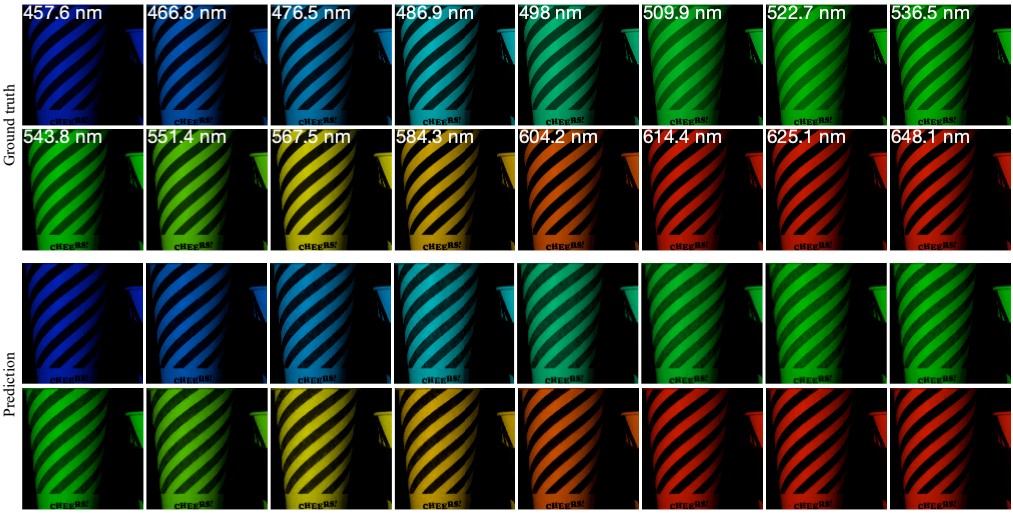

Figure 22: 1024×1024×28 high-resolution HSI reconstruction (scene 6): Sixteen spectral channels of ground truth HSI (upper two rows) and corresponding reconstructive results by SRN v2 model (lower two rows). Better visualization performance by zooming in.

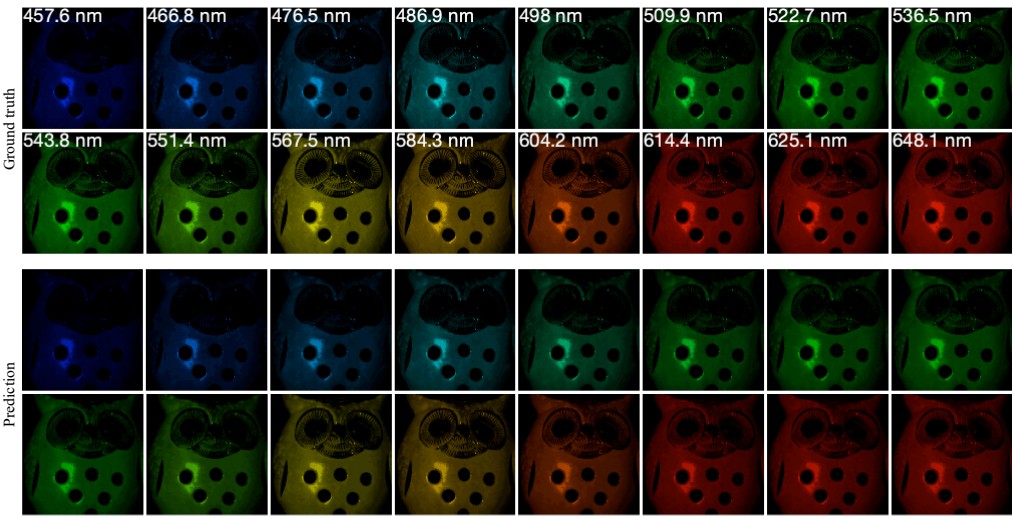

Figure 23: 1024×1024×28 high-resolution HSI reconstruction (scene 7): Sixteen spectral channels of ground truth HSI (upper two rows) and corresponding reconstructive results by SRN v2 model (lower two rows). Better visualization performance by zooming in.

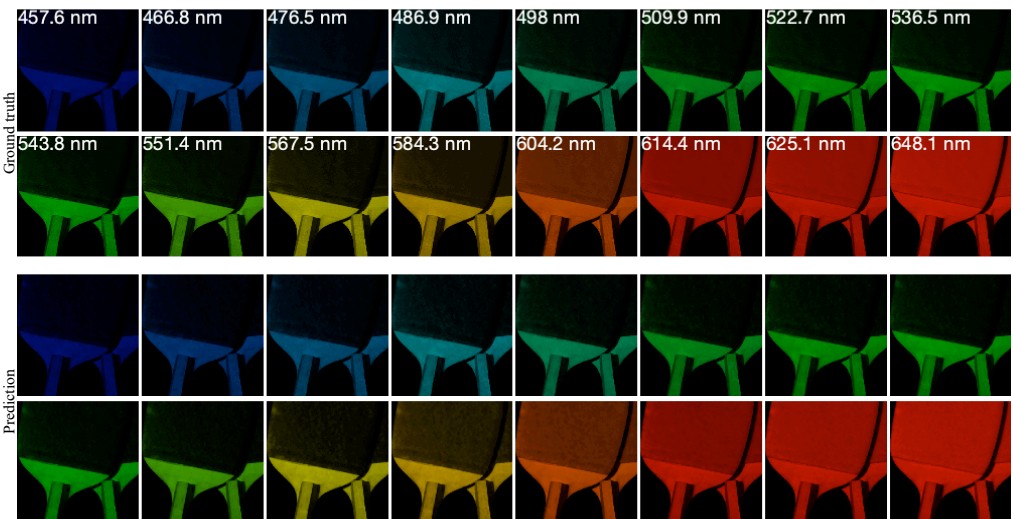

Figure 24: 1024×1024×28 high-resolution HSI reconstruction (scene 8): Sixteen spectral channels of ground truth HSI (upper two rows) and corresponding reconstructive results by SRN v2 model (lower two rows). Better visualization performance by zooming in.

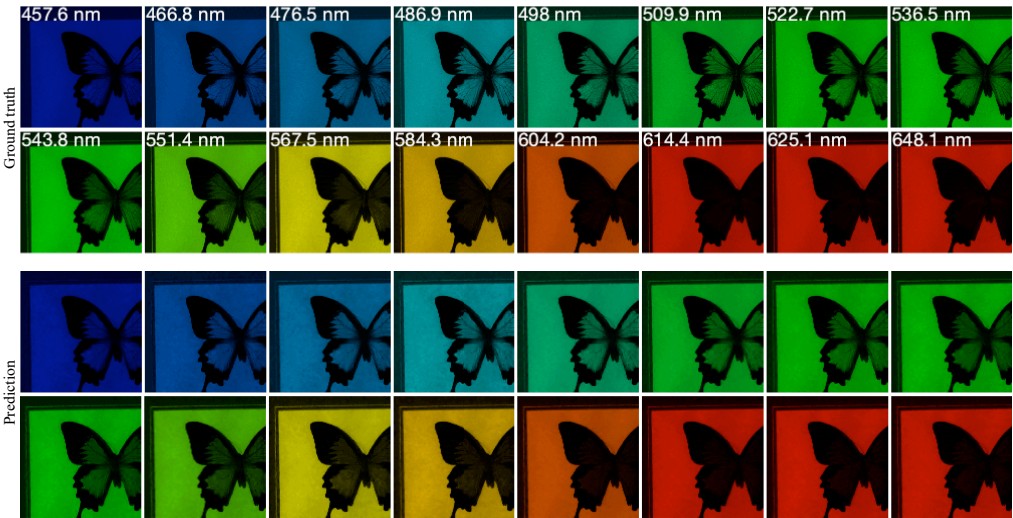

Figure 25: 1024×1024×28 high-resolution HSI reconstruction (scene 15): Sixteen spectral channels of ground truth HSI (upper two rows) and corresponding reconstructive results by SRN v2 model (lower two rows). Better visualization performance by zooming in.

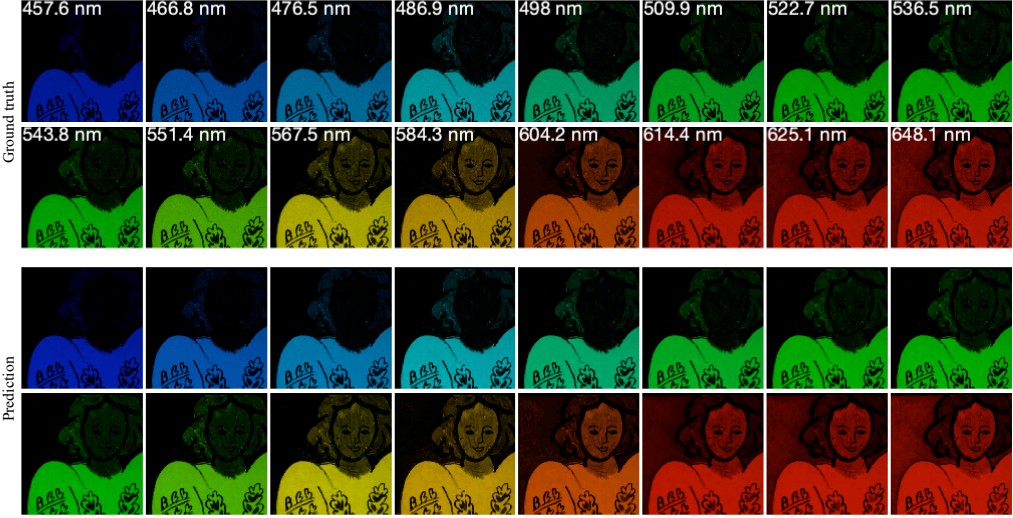

Figure 26: 1024×1024×28 high-resolution HSI reconstruction (scene 17): Sixteen spectral channels of ground truth HSI (upper two rows) and corresponding reconstructive results by SRN v2 model (lower two rows). Better visualization performance by zooming in.

