# OpenReview forum: "A NEW BACKBONE FOR HYPERSPECTRAL IMAGE RECONSTRUCTION"
_ICLR.cc/2022/Conference — ICLR 2022 Submitted_

### Official Review · Reviewer_gU7n · 2021-10-31

**Correctness:** 2
**Technical Novelty And Significance:** 3
**Empirical Novelty And Significance:** Not applicable
**Recommendation:** 5
**Confidence:** 4

**Main Review:**

Pros:
The writing of this paper is beautiful and informative.
The performance of the proposed method exceeds SOTA.
Demonstration of real hardware experiments is a plus.

Cons
About network position. In my understanding, new backbones should and could be integrated into other reconstruction networks, such as TSA-net, GSM-net, GAP-net, by replacing the U-Net in these networks. The backbone net should be plug-and-play and compatible with previous reconstruction networks. But, in this paper, the backbone is treated as a new reconstruction network, and be parallel with previous reconstruction networks. Further, the new backbone should be tested on other problems, such as video compressive imaging, image compressive sensing to demonstrate it is a backbone and not a new reconstruction method.
About analysis and motivation. There should be more analysis and motivation on the design principles. It is not clear why the proposed backbone can achieve such superior performance. There is some depiction about the receptive field, but I think RF is not the key issue, since networks with small or large RFs are easy to design and not all these networks could achieve such performance. It should demonstrate why the proposed network is the specific one that others can not replace.

Need Clarity
In the first paragraph in Sec. 3.2, it seems other methods use the measurement y during reconstruction, but the proposed method only uses F_Y. I think y and F_Y are different for HSI reconstruction. Please clarify these differences.
Eq. (1), when shifting signal frames of different wavelengths, how to deal with the boundary? Does it have an influence on real hardware experiments?
Figure. 6, spectral curves. I notice that the curves between Ours and References have a big deviation. Please check if something wrong.



**Summary Of The Paper:**

This paper proposes a new backbone for hyperspectral image reconstruction for CASSI system. The novel thing, as claimed, is a new network that can achieve high reconstruction fidelity with high efficiency.

**Summary Of The Review:**

I have concerns about the motivation of network design, together with the position of the proposed network.

---

### Official Review · Reviewer_ms6v · 2021-11-03

**Correctness:** 2
**Technical Novelty And Significance:** 1
**Empirical Novelty And Significance:** 3
**Recommendation:** 3
**Confidence:** 4

**Main Review:**

Strengthes:
- This work proposes a backbone network working very well for snapshot compressive hyperspectral imaging and achieved SOTA performance in a number of related benchmarks over recent related works that were presented at CVPR'21, ECCV'20.

Weaknesses:
- The main idea of the proposed method is to use skip connections between blocks and groups of blocks. However, there have been similar works on using multiple, repeated skip connections for backbone networks such as
[Huang CVPR 2017] Densely Connected Convolutional Networks
[Zhang ECCV 2018] Image Super-Resolution Using Very Deep Residual Channel Attention Networks
The proposed method and the above works (and their related works - they have been cited a lot) seem to share common ingredients and this work should address it properly to clearly show the contribution of the proposed backbone. Comparing with U-Net seems to have too limited scope only for hyperspectral community.
- This work says "We argue that in low-level regression tasks, its unnecessary for the highest RF to cover the whole image but is supposed to be large enough to capture the neighboring for the estimated pixel." However, compressive sensing matrix M generates samples by summing over the whole image domain with proper weights (0 or positive values). In other words, narrowing down the RF can potentially miss the encoded information in CASSI and I believe that that's why others in HSI used U-Net instead. Thus, if this argument is true, then this part should be carefully written again with proper arguments. Even though there was no issue in the experimental results, note that the dataset was not that large, so it is still possible to have good results in limited dataset with theoretical flaw to miss some information.
- For "EFFICIENCY MAXIMIZATION", this work sed a number of technics such as downsampling, upsampling and pixelshuffling. However, these technics are common in other image recovery domains such as super resolution as also indicated in this work.

Comments:
- This work says "As a result, a dimensional-expansion mapping function (2D to 3D) is required, for which reason such a mapping relationship approximation is deemed to be much harder than general image regression tasks". I am not sure if this claim is true. There are a number of ways to define how hard the inverse problem is and there are a number of general image regression tasks with very ill-posed forward model.
- It is not easy to agree with the motivations in section 2. For Proper Data Volume, it says "simpler model might be more promising." This paragraph argued with VC dimension, but there is no analysis on VC dimension to support this motivation. For Ideal reconstruction rate, there is also possibility to simply use non-compressed samples over CASSI for real-time applications. What are the cases of using CASSI with real-time applications? Please consider revising this part with more convincing details and with removing some parts that can not be supported later by experiments.
- It says "we posit the possibility of network construction with minimum inductive bias". How can we know that it is true? Any supporting data for it? Small RF seems to introduce bias by removing some measurements to use and small training / evaluation datasets seems also to be an issue for it.
- Please comment on training time.


**Summary Of The Paper:**

This work proposes a novel backbone network for hyperspectral image reconstruction for snapshot compressive imaging (CASSI). The key structure of the proposed method is illustrated in Figure 3, essentially using skip connections between Conv-ReLU-Conv blocks repeatedly. A number of techniques such as downsampling, upsampling, pixelshuffling were employed to maximize efficiency. The proposed method achieved SOTA performance in the given CASSI reconstruction tasks over recent works with small network size and fast computation.

**Summary Of The Review:**

This work may be a nice contribution to the field of hyperspectral imaging, but it is not easy to see this work as a good contribution for ICLR; using dense skip connections is not new even though the way of connecting them is different from others, using pixelshuffling is common in super resolution field, and narrowing down the RF does not seem to be theoretically solid since it is then possible to miss some information.

---

### Official Review · Reviewer_NYcV · 2021-11-03

**Correctness:** 3
**Technical Novelty And Significance:** 2
**Empirical Novelty And Significance:** 3
**Recommendation:** 5
**Confidence:** 3

**Main Review:**

Strengths:

- This paper address an important problem in hyperspectral image processing, i.e. reconstruction from single images. The proposed method is well motivated with the drawbacks of the current methods clearly described.

- The proposed method has a simple network structure, therefore, can be easily re-implemented. The network structure has well addressed the motivation of this paper.

- The proposed method is very effective. It produces higher PSNR and SSIM than previous approaches. It also significantly reduces the number of parameters and flops, therefore, is efficient and affordable when facing video processing tasks.

Weakness:

- Hyperspectral image reconstruction has attracted more and more attention in the past several years. Many papers do not study the mechanism behind the imaging process, the property of hyperspectral images and how to make the reconstructed images really useful, instead focus on adopting some deep learning techniques to learn the mapping from single-shot measurement to multiple channel image. This paper is one of them, and is not exciting with so many papers doing the same thing. I'd rather want to see a paper that works on reconstructing images with a wider spectral range and a much higher number of bands, more complex scenes consisting of multiple objects, and addressing a real-world problem such as distinguishing the fine differences of similar materials. These are the reason that makes hyperspectral image useful and the reconstruction of hyperspectral image meaningful.

- The novel contribution on network mechanism and image representation learning is low. The design of the structure seems to be empirical. Although the proposed model is simple, its rationale has not been clearly explained, especially why and how it can produce high reconstruction quality.

- The presentation of the paper shall be improved. There are some typos and errors that shall be fixed, e.g. "nad", "Fig. 6 (a)" (there is no Fig. 6 (a)), and "Tab ??".

More comments:

1) When introducing U-net, the properties of medical images are described, which is not closely relevant to this paper.
2) Discussion on the receptive field is not convincing. There are differences between medical image and hyperspectral image, but such difference is simply on the data and can be easily solved by introducing different training samples. It is not related to the network model. The authors commented that "medical image processing task is always more locally focused while the HSI reconstruction tends to be globally comprehensive". However, isn't HSI reconstruction also focus on local details? This has been highlighted in the experimental results, e.g. Figure 5 that shows more details of the scene?
3) In evaluating the quality of reconstructed spectral responses, has the authors considered the spectral angle distance between the estimated signal and the ground truth? This is more commonly used in remote sensing for quantitative evaluation.
4) Since the depth of network is highly important, an analysis on the impact of K shall be given.
5) "Within the chosen region (green patch), our reconstructed pixels yield the highest a correlation with the reference given by spectrometer, indicating the effectiveness of our method within waveband of green, i.e., 500nm-565nm". Why green patch? How is the overall quality of the reconstructed spectral responses?
6) Section 3.1 is highly similar to the description on CASSI in some published papers. Even if the papers may be written by the same authors, such similarity shall be reduced or avoided.

**Summary Of The Paper:**

This paper introduces a hyperspectral image reconstruction method that aims to produce an image with multiple channels from a single-shot measurement with high reconstruction quality and efficiency. This is achieved through the development of a simple reconstruction network that adopts some tricks such as spatial and spectral invariant learning and identity connection. The proposed method was tested on CAVE and KAIST datasets, and was compared with several state-of-the-art methods. The results show clear advantages of the proposed method.

**Summary Of The Review:**

This paper introduces a simple but effective and efficient method for hyperspectral image reconstruction. However, the novelty and significance of the contribution of the paper are not high.

---

### Official Review · Reviewer_dXDL · 2021-11-23

**Correctness:** 3
**Technical Novelty And Significance:** 3
**Empirical Novelty And Significance:** 3
**Recommendation:** 6
**Confidence:** 3

**Main Review:**

This paper proposed an intersting points, that is to say, in lowlevel regression tasks, its unnecessary for the highest RF to cover the whole image but is supposed to be large enough to capture the neighboring for the estimated pixel. The authors also compared the proposed method to the U-Net regarding the receptive field. In fact, the authors proved in the experimental section that the proposed architecture can achieve better performance compared to Unet. I'm still confusing about the conclusion that enough RF to capture the neighboring for the estimated pixel is better for the hightest RF to cover the whole image. I suggest an analysis study should be presented about the RF and the proposed method.
The authors proposed three kinds of variants on the bisis of the proposed backbone. The parameters of different variants stay similar, but the flops changes significantly. Furthermore, the performance of these three variants are also of big diversity. I'm wondering what is the main factor to increase the peformance. The flops indicates the training efficiency. Why not point out the training time to show a direct efficiency of different methods?
The paper described that adjacent spectral channels tend to be more correlated to each other in general HSIs. What is the reference?
How many training samples and total epochs utilized in the training stage for simulated and real experiments?
As shown in Fig. 5, the proposed method introduced the artifact at the right top of the image, compared to that of GAP-net.

**Summary Of The Paper:**

Different from the previous works about Unet structure for HSI reconstruction, this paper introduces a new backbone, which is efficient and lightweight. The experiments also demosntrate that the proposed model can significantly outperfom the previous works in quantitative evaluation results. From the above summary, this paper should be of significance to the application society of CASSI imaging reconstruction.

**Summary Of The Review:**

The experimental results of the proposed method have largely outperformed baseline methods.

---

### Decision · Program_Chairs · 2022-01-20

**Decision:**

Reject

**Comment:**

The paper proposes a new neural network architecture for hyperspectral image reconstruction. The paper received borderline/negative reviews. Significant concerns were raised about the novelty and significance of the contribution. Unfortunately, the authors did not upload a rebuttal, preventing the reviewers from changing their opinion about the paper. There is therefore no reason to overturn their recommendation.